# INNATECODER: LEARNING PROGRAMMATIC OPTIONS WITH FOUNDATION MODELS

## ABSTRACT

Outside of transfer learning settings, reinforcement learning agents start their learning process from a clean slate. As a result, such agents have to go through a slow process to learn even the most obvious skills required to solve a problem. In this paper, we present INNATECODER, a system that leverages human knowledge encoded in foundation models to provide programmatic policies that encode "innate skills" in the form of temporally extended actions, or options. In contrast to existing approaches to learning options, INNATECODER learns them from the general human knowledge encoded in foundation models in a zero-shot setting, and not from the knowledge the agent gains by interacting with the environment. Then, INNATECODER searches for a programmatic policy by combining the programs encoding these options into larger and more complex programs. We hypothesized that INNATECODER's way of learning and using options could improve the sampling efficiency of current methods for learning programmatic policies. We evaluated our hypothesis in MicroRTS and Karel the Robot, two challenging domains. Empirical results support our hypothesis, since they show that INNATECODER is more sample efficient than versions of the system that do not use options or learn the options from experience. The policies INNATECODER learns are competitive and often outperform current state-of-the-art agents in both domains.

## 1 INTRODUCTION

Outside of transfer learning settings, deep reinforcement learning (DRL) agents begin their learning process with randomly initialized neural networks. As a result, DRL agents must learn from scratch even the most basic skills required to solve a problem. In this paper, we harness the general human knowledge encoded in foundation models to endow agents with helpful skills before they even start interacting with the environment. This is achieved by using programmatic representations of policies (Trivedi et al., 2021)—programs written in a domain-specific language encoding policies—and the foundation models' ability to write computer programs. Depending on the language used, programmatic policies were shown to generalize better to unseen scenarios (Inala et al., 2020) and to be human-interpretable (Verma et al., 2018; Bastani et al., 2018). In addition to these advantages and loosely inspired by the innate abilities of animals (Tinbergen, 1951), we show that programmatic representations of policies allow us to harness helpful "innate skills" from foundation models.

Given a natural-language description of the problem that the agent needs to learn to solve, our system, which we call INNATECODER, queries a foundation model for programs that encode policies to solve the problem. Although the programs the model generates are unlikely to encode policies that solve the problem, we hypothesize that the set of sub-programs we obtain from these programs can encode helpful temporally extended actions, or options (Sutton et al., 1999). We consider options as functions the agent can call and that will tell it how to act for a number of steps (Precup et al., 1998).

Options can ease the agent's learning process in different ways. For example, options can allow the agent to better explore the problem space (Machado et al., 2017; Bellemare et al., 2020) or can transfer knowledge between different tasks (Konidaris & Barto, 2007). In this paper, we present a novel way of learning options with foundation models. We also present a novel way of using the learned options, which is inspired by recent work on learning semantic spaces of programming languages (Moraes & Lelis, 2024). We leverage the compositional nature of the programmatic options we harness from a foundation model to learn the underlying semantic space of the programming

$\rho :=$ `if` $h$ `then` $a$

$h :=$ `frontIsClear` | `markersPresent`

$a :=$ `move` | `putMarker` | `pickMarker`

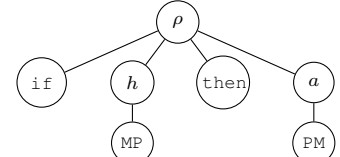

Figure 1: Left: The context-free grammar specifying a simplified version of the domain-specific language for Karel the Robot, a benchmark we use in our experiments. Right: the abstract syntax tree for `if markersPresent then pickMarker`. In the tree, `MP` and `PM` stand for `markersPresent` and `pickMarker`, respectively. Karel is a robot acting on a grid, where it needs to accomplish tasks such as collecting and placing markers on different locations of the grid. In this program, Karel will pick up a marker if one is present in its current location on the grid.

language that defines the agent's hypothesis space. In the semantic space, neighbor programs encode similar but different agent behavior, which is a desirable property when inducing spaces conducive to algorithms searching for programmatic policies (Trivedi et al., 2021). The semantic space is approximated by ensuring that neighboring policies differ in term of one sub-policy from the set of options. Instead of searching in the space of programs induced by the syntax of the language (Koza, 1992), INNATECODER searches in the space of semantically different programs induced by options. In contrast with previous methods that can benefit from up to hundreds of options (Eysenbach et al., 2019), INNATECODER's use of programmatic options allows it to benefit from thousands of them.

INNATECODER's approach to harnessing options from foundation models contrasts with previous approaches to automatically learning them, e.g., (Tessler et al., 2017; Bacon et al., 2017; Igl et al., 2020; Klissarov & Machado, 2023). This is because options are harnessed from the general knowledge encoded in a foundation model, as opposed to the knowledge the agent gains by interacting with the environment. This zero-shot approach to learning options is enabled by the use of a domain-specific language to bridge the gap between the high-level knowledge encoded in foundation models and the low-level knowledge required at the sensorimotor control level of the agent (Klissarov et al., 2024). For example, foundation models trained on Internet data likely encode the knowledge that, to win a match of a real-time strategy game, the player must collect resources and build structures, which will allow for the training of the units needed to win the game. However, the model cannot issue low-level actions in real time to control dozens of units to accomplish this plan. INNATECODER bridges this gap by distilling the knowledge of the model into options that can be executed in real time.

We evaluated our hypothesis that foundation models can generate helpful programmatic options in the domains of MicroRTS, a challenging real-time strategy game (Ontañón, 2017), and Karel the Robot (Pattis, 1994), which has been used as a benchmark for program synthesis and reinforcement learning algorithms (Bunel et al., 2018; Chen et al., 2018; Shin et al., 2018; Trivedi et al., 2021). The results in both domains support our hypothesis, since INNATECODER was more sample-efficient than versions of the system that do not use options or learn options from experience. We also show that the policies INNATECODER learns are competitive and often outperform the current state-of-the-art algorithms. INNATECODER is inexpensive because it uses the foundation model a small number of times as a pre-processing step, making it an accessible system to smaller labs and companies.

## 2 PROBLEM DEFINITION

We consider sequential decision-making problems that can be formulated as Markov decision processes (MDPs) $(S, A, p, r, \mu, \gamma)$. Here, $S$ represents the set of states and $A$ is the set of actions. The function $p(s_{t+1}|s_t, a_t)$ is the transition model, which gives the probability of reaching state $s_{t+1}$ given that the agent is in $s_t$ and takes action $a_t$ at time step $t$. The agent observes a reward value of $R_{t+1}$ when transitioning from $s_t$ to $s_{t+1}$. The reward value the agent observes is returned by the function $r$. $\mu$ is the distribution of the initial states of the MDP; states sampled from $\mu$ are denoted $s_0$. $\gamma$ in $[0, 1]$ is the discount factor. A policy $\pi$ is a function that receives a state $s$ and returns a probability distribution over actions available at $s$. The goal is to learn a policy $\pi$ that maximizes the expected sum of discounted rewards for $\pi$ starting in an $s_0$: $\mathbb{E}_{\pi, p, \mu}[\sum_{k=0}^{\infty} \gamma^k R_{k+1}]$. $V^{\pi}(s) = \mathbb{E}_{p, \pi}[\sum_{k=0}^{\infty} \gamma^k R_{k+1}|s_0 = s]$ is the value function, which measures the expected return

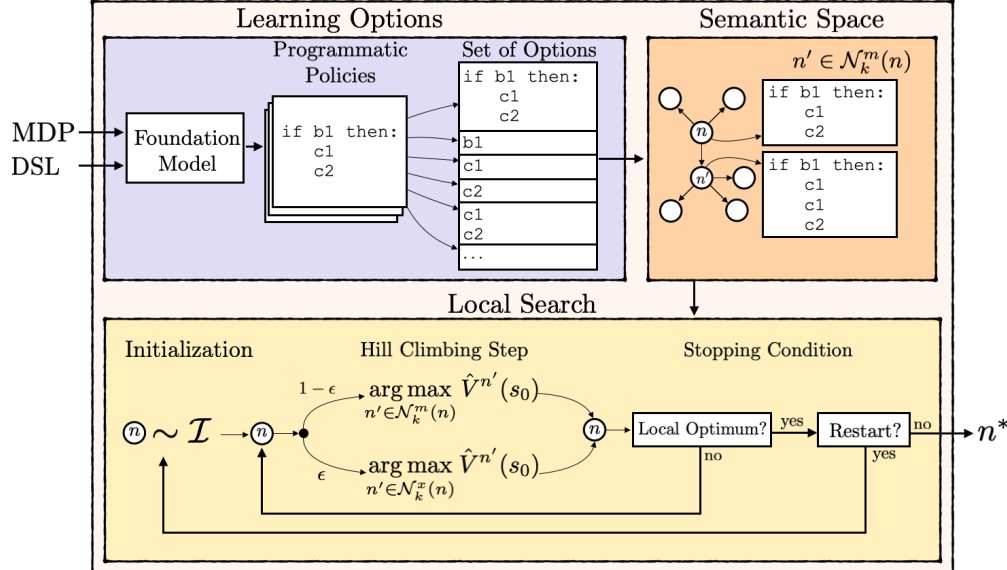

Figure 2: Schematic view of INNATECODER, with three parts. The "Learning Options" component harnesses options from a foundation model from a natural language description of the MDP and a Backus-Naur form description of the DSL. The model generates a set of programmatic policies that are broken down into a set of options (Section 3.1). The "Semantic Space" component uses the options to approximate the semantic space of the DSL (Section 3.2). The "Local Search" component searches in a mixture of the syntax and semantic spaces for a programmatic policy $n^*$ (Section 3.3).

when the agent follows the policy $\pi$ starting from the state $s$. In this work, we approximate the value function of a policy $\pi$ and state $s$ with Monte Carlo roll-outs and denote the approximation as $\hat{V}^\pi(s)$.

We consider programmatic representations of policies, which are policies written in a domain-specific language (DSL). The set of programs a DSL accepts is defined through a context-free grammar $(M, N, R, I)$, where $M$, $N$, $R$, and $S$ are the sets of non-terminals, terminals, the production rules, and the grammar's initial symbol, respectively. Figure 1 shows a DSL for a simplified version of the language we use in our experiments for Karel the Robot (the complete DSL is shown in Appendix H.1). In this DSL, the set $M$ is composed of symbols $\rho$, $h$, $a$, while the set $N$ includes the symbols if, frontIsClear, markersPresent, move, putMarker, pickMarker. $R$ are the production rules (e.g., $h \rightarrow$ frontIsClear), and $\rho$ is the initial symbol. We denote programmatic policies with letters $p$ and $n$ and their variations such as $n'$ and $n^*$.

We represent programs as abstract syntax trees (AST), where each node $n$ and its children represent a production rule if $n$ represents a non-terminal symbol. For example, the root of the tree in Figure 1, which represents the non-terminal $\rho$; node $\rho$ and its children represent the production rule $\rho \rightarrow$ if $h$ then $a$. Leaf nodes in the AST represent terminal symbols. Figure 1 shows an example of an AST for the program if markersPresent then pickMarker. A DSL $D$ defines the possibly infinite space of programs $[\![D]\!]$, where in our case each program $p$ in $[\![D]\!]$ represents a policy.

Given a domain-specific language $D$, our task is to find a programmatic policy $p \in [\![D]\!]$ that maximizes the expected sum of discounted rewards for a given MDP.

## 3 INNATECODER

Figure 2 shows a schematic view of INNATECODER, which receives an MDP and a DSL and returns a programmatic policy, denoted $n^*$. INNATECODER is composed of three components: one for learning options, another that uses learned options to induce an approximation of the semantic space of the DSL, and a component to search in such a space. In this section, we explain the three components.

## 3.1 Learning Options

An option is a program encoding a policy that the agent can invoke at specific states. Once invoked, the program tells the agent what to do for a number of steps. Once completed, the option "returns" the control back to the agent. An option $\omega$ is defined with a tuple $(I_\omega, \pi_\omega, T_\omega)$, where $I_\omega$ is the set of states in which the option can be initiated; $\pi_\omega$ is the policy the agent follows once $\omega$ starts; $T_\omega$ is a function that returns the probability in which $\omega$ terminates at a given state $s_t$. INNATECODER uses a foundation model to learn programs, written in a given DSL, encoding options. The programs receive a state of the MDP and return the action the agent should take at that state, thus encoding $\pi_\omega$.

We assume that the set $I_\omega$ is the set $S$ of all states of the MDP, which means that the program can be invoked in any state. However, note that the program may not return any actions in a given state $s$, which is equivalent to $s$ not being in $I_\omega$. For example, "if $b_1$ then $c_1$" returns the action given in $c_1$ only if condition $b_1$ is satisfied in the state in which the option was queried; it returns no action for states that do not satisfy $b_1$. The option termination criterion $T_\omega$ is also determined by the program; the option terminates when the program terminates. This termination criterion means that, depending on the DSL, options have an internal state, representing the line in the program in which the execution will continue the next time the agent interacts with the environment. For example, if the option "$c_1$ $c_2$" is invoked for state $s_t$ and $c_1$ returns an action, then the agent's action in $s_{t+1}$ is determined by $c_2$.

Programmatic options are harnessed from a foundation model as follows. We provide a natural language description of the MDP and the Backus-Naur form of the DSL to the model. The model then provides a set of $m$ programs written in the DSL encoding policies for the MDP. While it is unlikely that the model can provide policies that can maximize the expected return for any MDP of interest, we hypothesize that these programmatic policies can be broken up into sub-programs that can encode helpful agent behavior. Each program $p$ is broken up into one sub-program for each sub-tree rooted at a non-terminal symbol in the AST of $p$. For example, for the program "if $b_1$ then $c_1$ $c_2$" we obtain the sub-programs "if $b_1$ then $c_1$ $c_2$", "$b_1$", "$c_1$", "$c_2$", and "$c_1$ $c_2$". These sub-programs form a set of options $O$, which INNATECODER uses to approximate the semantic space of the DSL. Note that this set of options is generated zero-shot, before the agent starts interacting with the environment.

## 3.2 Approximating the Semantic Space with Options

Methods for searching for programmatic policies traditionally search in the space of programs defined by the context-free grammar of the DSL (Koza, 1992; Verma et al., 2018; Carvalho et al., 2024). We refer to this type of space as the syntax space, since it is based on the syntax of the language.

**Definition 1 (Syntax Space)** *The syntax space of a DSL $D$ is defined by $(D, \mathcal{N}_k^x, \mathcal{I}, \mathcal{E})$. With $[\![D]\!]$ defining the set of candidate programs, or solutions, $\mathcal{N}_k^x$ ($x$ is for "syntax") is the syntax neighborhood function that receives a candidate and returns $k$ candidates from $[\![D]\!]$. $\mathcal{I}$ is the distribution of initial candidates. Finally, $\mathcal{E}$ is the evaluation function, which receives a candidate in $[\![D]\!]$ and returns a value in $\mathbb{R}$.*

A common way of defining the distribution of initial candidates $\mathcal{I}$ is through a procedure that starts with a string that is the initial symbol of the grammar and iteratively, and uniformly at random, samples a production rule to replace a non-terminal symbol in the string. In the example of Figure 1, we replace the initial symbol $I$ with "if($B$) then $C$" with probability $1.0$, since this is the only rule available; then, $B$ is replaced with either "$b_1$" or "$b_2$" with probability $0.5$ each. This iterative process stops once the string contains only terminal symbols. If a probabilistic context-free grammar is available, then the distribution $\mathcal{I}$ can be defined through the same process, but using the probabilities from the grammar as opposed to a uniform distribution over production rules (Trivedi et al., 2021).

The syntax neighborhood function $\mathcal{N}_k^x$ defines the structure of the search space, as it determines the set of candidate solutions (programs that encode a policy) that the search procedure can evaluate from a given candidate $n$. Given a candidate $n$, $\mathcal{N}_k^x(n)$ returns a set of $k$ neighbors of $n$. These candidates are generated by selecting uniformly at random a node that represents a non-terminal symbol in the AST of $n$. Then, the sub-tree rooted at the selected node is replaced by another sub-tree generated using the process described for $\mathcal{I}$, but starting at the non-terminal symbol the node represents. This process of replacing a sub-tree in $n$ is repeated $k$ times, to generate $k$ possibly different neighbors of

$n$. Finally, $\mathcal{E}$ is an approximation of the value function of the policy encoded in $n$ from a set of initial states $s_0$, $\hat{V}^n(s_0)$; we obtain $\hat{V}^n(s_0)$ by averaging the returns after rolling $n$ out from states $s_0$.

Moraes & Lelis (2024) showed that searching in the syntax space can be inefficient because often the neighbors $n'$ of a candidate $n$ encode policies that are semantically identical to $n$; the programs differ in terms of syntax, but encode exactly the same agent behavior. As a result, the search process wastes time evaluating the same agent behavior. Their solution is to approximate the underlying semantic space of the language, where neighbor programs are similar in terms of syntax, but are likely to differ in terms of behavior. In their setting, the agent learns programs for a set of tasks, which are used to induce the semantic space, and the induced space is used in downstream tasks. INNATECODER overcomes the requirement to operate on a stream of problems by using a foundation model to learn the options in a zero-shot setting. A semantic space is defined as follows.

**Definition 2 (Semantic Space)** *The semantic space of a DSL $D$ is defined by $(D, \mathcal{N}_k^m, \mathcal{I}, \mathcal{E})$, where $\mathcal{I}$ and $\mathcal{E}$ are identical to the syntax space (Definition 1). The function $\mathcal{N}_k^m$ ($m$ is for "semantics") is a semantic neighborhood function that also receives a candidate and returns $k$ candidates from $[\![D]\!]$.*

We define the function $\mathcal{N}_k^m$ with a set of options $\Omega$, where each option $\omega$ in $\Omega$ represents a different agent behavior. A neighbor of candidate $n$ is then obtained by selecting, uniformly at random, a node $c$ in the AST of $n$ that represents a non-terminal symbol. Then, we replace the sub-tree rooted at $c$ with the AST of an option $\omega$ in $\Omega$. The option $\omega$ is selected, uniformly at random, among those in $\Omega$ whose AST root represents the same non-terminal symbol $c$ represents. By matching the non-terminal symbols when selecting $\omega$, we match $\omega$ with the type of the sub-tree that is removed from $n$. Similarly to $\mathcal{N}_k^x$, $\mathcal{N}_k^m$ generates $k$ possibly different neighbors by repeating this process $k$ times.

The intuition for requiring the options in $\Omega$ to encode different agent behaviors is to increase the chance of seeing neighbors with different behaviors. For example, if many of the options in $\Omega$ encoded exactly the same behavior, then the chances of all neighbors of a program also encoding the same behavior would be higher, which is wasteful from a sample efficiency perspective. Next, we describe how we obtain $\Omega$ from the set of options $O$ we harnessed from a foundation model.

We filter the set of options $O$ harnessed from the foundation model into a set $\Omega$ of behaviorally different options by having the agent interact with the environment, as described in previous work (Moraes & Lelis, 2024). Assuming an MDP with discrete actions, each option in $O$ is evaluated in an ordered set of states $\mathcal{S}$ of the MDP. This set $\mathcal{S}$ is obtained by rolling out all options $o \in O$ once, from an initial state $s_0$ sampled from $\mu$. The states $s$ observed in this process form $\mathcal{S}$. Then, every option is invoked for each state in $\mathcal{S}$, thus forming an *action signature* $A_o$ for each $o$. An action signature is a vector with one action for each state in $\mathcal{S}$, where the $i$-th entry of $A_o$ corresponds to the action $o$ returns to the $i$-th state in $\mathcal{S}$. The set of options $\Omega$ is given by one option for each observed $A_o$. If multiple options have the same signature, we arbitrarily select one of them.

The number of samples required to filter the set of options into a set of options with different behaviors is negligible: it uses less than 1% of the computation in our experiments. Programs that cannot be rolled out (e.g., Boolean expressions and do not issue actions) are not included in the set of options.

## 3.3 SEARCHING IN SEMANTIC SPACE

INNATECODER uses stochastic hill-climbing (SHC) to search in the semantic space of a given DSL $D$ for a policy that maximizes the agent's return. SHC starts its search by sampling a candidate program $n$ from $\mathcal{I}$. In every iteration, SHC evaluates all $k$ neighbors of $n$ in terms of their $\mathcal{E}$-value. The search then moves on to the best neighbor of $n$ in terms of $\mathcal{E}$, and this process is repeated from there. SHC stops if none of the neighbors has an $\mathcal{E}$-value that is better than the current candidate, that is, it reaches a local optimum. SHC uses a restarting strategy: once SHC reaches a local optimum, if SHC has not yet exhausted is search budget, it restarts from another initial candidate sampled from $\mathcal{I}$. SHC returns the best solution, denoted $n^*$, encountered in all restarts of the search.

INNATECODER does not search solely in the semantic space, but mixes both syntax and semantic spaces in the search. This is because the set of options might cover only a part of the space of programs the DSL induces. To guarantee that INNATECODER can access all programs in $[\![D]\!]$, with probability $\epsilon$, SHC uses the syntax neighborhood function in the search, and with probability $1 - \epsilon$, it

uses the semantic one. We use $\epsilon = 0.4$ in our experiments. We chose this value because it performed better in preliminary experiments than the value of $0.2$ used in previous work (Moraes & Lelis, 2024).

Although other local search algorithms could be used with INNATECODER, such as Simulated Annealing (Kirkpatrick et al., 1983; Husien & Schewe, 2016), we use SHC because previous work showed that it performs well in our test domains (Moraes et al., 2023; Carvalho et al., 2024).

## 4 EMPIRICAL EVALUATION

Although foundation models are unlikely to generate programs that encode policies for fully solving MDPs, we hypothesize that the programs they generate can be broken up into smaller programs that serve as helpful options. We evaluated the usefulness of these programmatic options by measuring the sampling efficiency of search algorithms searching in the semantic spaces induced by them. We evaluate INNATECODER on MicroRTS (Ontañón, 2017) and Karel the Robot (Pattis, 1994).

**MicroRTS**  MicroRTS is a real-time strategy game that requires the agent to control dozens of units in real-time, thus making it impractical to use foundation models to decide on agent actions directly. We use the following maps from the MicroRTS repository,[1] with the map size in brackets: NoWhereToRun ($9 \times 8$), basesWorkers ($24 \times 24$), and BWDistantResources ($32 \times 32$), and BloodBath ($64 \times 64$). We use these maps because they differ in size and structure. Since MicroRTS is a multi-agent problem, we use 2L, a self-play algorithm, to learn programmatic policies (Moraes et al., 2023). MicroRTS is not a symmetric game and the outcome of the game depends on the starting location of the players. To ensure fairness, each pair of policies plays two matches on each map, so that each player can start at each of the two initial locations; the results are then averaged out of these two runs. In the context of 2L, INNATECODER is required to solve an MDP in every iteration of self-play (see Appendix M). We use a new version of the MicroLanguage as the DSL (Mariño et al., 2021). The language offers specialized functions and an action-prioritization scheme through for-loops, where nested for-loops allow for higher priority of actions. We provide a detailed explanation of the MicroLanguage, as well as images of the maps used, in Appendices H.2 and K, respectively.

**Karel**  Karel the Robot is an environment originally created for teaching people how to write computer programs, which has later been used as a benchmark domain for reinforcement learning algorithms (Trivedi et al., 2021). Karel is a robot interacting with a grid-world, where it can collect markers and place markers. We use the following Karel problems, which were designed in previous works (Trivedi et al., 2021; Liu et al., 2023b): StairClimber, FourCorners, TopOff, Maze, CleanHouse, Harvester, DoorKey, OneStroke, Seeder, and Snake. The problems differ in terms of structure of the grids (e.g., where walls are located) and in terms of the task that Karel needs to accomplish. For example, in CleanHouse, Karel needs to collect all markers placed in the grid, while in TopOff it has to place a marker on top of all existing markers. The problem are described in Appendix I. We use the more difficult version of the environment known as "crashable" (Carvalho et al., 2024), where an episode terminates with a negative reward if Karel bumps into a wall. We use the same DSL used in previous work (Trivedi et al., 2021), which we describe in Appendix H.1.

**Baselines**  The current state-of-the-art methods for both MicroRTS and Karel use programmatic representations of policies, where the policies are written in the DSLs we use in our experiments (Moraes et al., 2023; Trivedi et al., 2021). Therefore, we focus on methods that use programmatic representations as baselines. However, we provide comparisons of INNATECODER with deep reinforcement learning baselines in Appendices B.1 (MicroRTS) and B.2 (Karel). For both MicroRTS and Karel we use SHC searching in the syntax space as a baseline, as it represents state-of-the-art performance in both domains (**SHC**). We also use two variants of INNATECODER where the options are learned without the help of a foundation model. These variants can be seen as implementations of the Library-Induced Semantic Spaces (LISS) (Moraes & Lelis, 2024) for a non-transfer learning setting. In the first variant, LISS learns the options as it learns how to solve the problem. In terms of the scheme shown in Figure 2, we skip the "Learning Options" step and build the set of options from the programs returned in every complete search of SHC. That is, when we reach the box "Restart?", we use the sub-programs of the best program encountered in that search to augment the set of options.

---

[1]https://github.com/Farama-Foundation/MicroRTS/

We call this baseline **LISS-o**, where "o" stands for "online". In the second variant, we sample programs from $\mathcal{I}$ and use their sub-programs to form the set of options. We call this baseline **LISS-r**, where "r" stands for "random". We also use the best program the foundation model generated out of all programs used to create the set of options as a baseline, which we call **FM**, which stands for "foundation model". LISS-o and LISS-r allow us to evaluate the effectiveness of learning options from a foundation model, while FM allows us to evaluate the foundation model as an alternative to solve the problem directly. We also use the Cross Entropy Method (CEM) operating in a learned latent space, which was shown to outperform DRL algorithms in all Karel tasks (Trivedi et al., 2021).

**Foundation Models** We use OpenAI's API for GPT 4o, whose training cut-off date is October 2023. We also perform tests, for MicroRTS, using the LLama 3.1 model with 405 billion parameters, whose training cut-off is December 2021. We used the GPT model in both MicroRTS and Karel experiments, while the Llama model was used in MicroRTS experiments. There were no MicroRTS programs available online prior to the Llama cut-off date, so the Llama evaluations on MicroRTS did not suffer from data leakage. The GPT model might have trained on the MicroRTS and Karel programs that were available online prior to its training cut-off date. We attempt to measure how much a possible data leakage can influence our results by using the FM baseline. If the model can simply retrieve the solutions seen in training, one would expect this baseline to perform well.

**Other Specifications** All experiments were run on 2.6 GHz CPUs with 12 GB of RAM. We use $k = 1,000$ in the neighborhood function. In MicroRTS, SHC is run with a restarting time limit of $2,000$ seconds for each self-play iteration. In Karel, since we are solving a single MDP, SHC restarts as many times as possible within the computational budget. For MicroRTS, we query the foundation models 120 times to generate the same number of programs; for Karel, we use 100 programs. We use the same number of programs as the LISS-r baseline. We perform 30 independent runs (seeds) of each system, including the generation of the programs by the foundation model.

**Metrics of Performance** For MicroRTS, performance is measured in terms of winning rate. The winning rate of a policy is computed for a set of opponent policies and is computed as follows: we sum the number of victories and half the number of draws and divide this sum by the total number of matches played (Ontañón, 2017). For Karel, performance is measured in terms of episodic return (Trivedi et al., 2021). We use prompts where we briefly describe each problem and provide a formal description of the DSL used. The prompts used in our experiments are in Appendices L (MicroRTS) and J (Karel). Both MicroRTS and Karel are deterministic, so the value of $\mathcal{E}$ for policies can be computed with a single roll-out. We report average performance and 95% confidence intervals.

**Efficiency Experiment** We verify the sampling efficiency of INNATECODER, LISS-o, LISS-r, and SHC. Similarly to previous work, we present learning curves, where for MicroRTS, we plot winning rate by the number of games played (Figure 3), and for Karel, we plot episodic return by the number of episodes (Figure 4). For MicroRTS, the winning rate is computed for a system by having the policy the system generated, after a given number of games played, play against the policies each of the other systems generated after the maximum number of games played (rightmost point of each plot).

**Competition Experiment** We evaluate INNATECODER against COAC, Mayari, and RAISocketAI, the winners of the previous three MicroRTS competitions. We randomly select 9 from the 30 programs generated in the "Efficiency Experiment" and evaluate them against the competition winners. We report the average results of the 9 programs against each opponent in the four maps we use.

**Size and Information Experiments** We also evaluate the effect of the size of the set of options on the sample efficiency of INNATECODER. We evaluated sets $\Omega$ with 300, 600, 1400, 5000, 7000 and 30000 options on the LetMeOut map ($16 \times 8$); all options were generated with the Llama 3.1 model. In Appendix C, we evaluate INNATECODER using prompts with more or less information and in Appendix D with GPT 3.5, to verify if performance decreases by using a smaller foundation model.

### 4.1 LEARNING CURVE RESULTS

Figures 3 and 4 show the learning curves for MicroRTS and Karel, respectively, where INNATECODER is denoted as IC-GPT or IC-Llama, depending on the model it uses to learn the programmatic options.

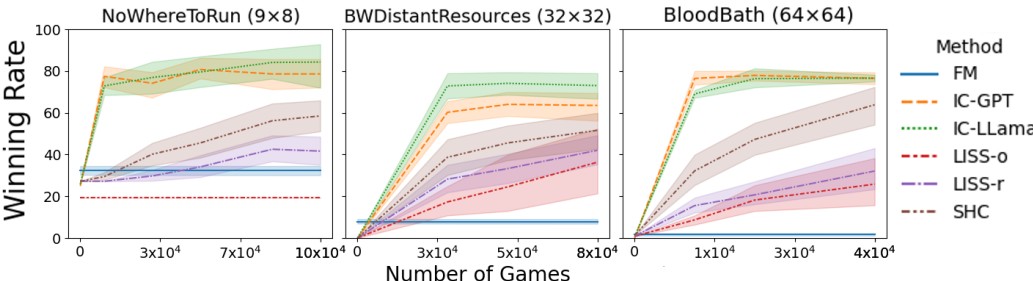

Figure 3: Winning rate (maximum is 100) per number of games played. The winning rate of the policies each system generates for a given number of games played is computed considering as opponents the policies all systems generate at the end of the learning process. The plots show the average winning rate of 30 independent runs (seeds) and the 95% confidence interval.

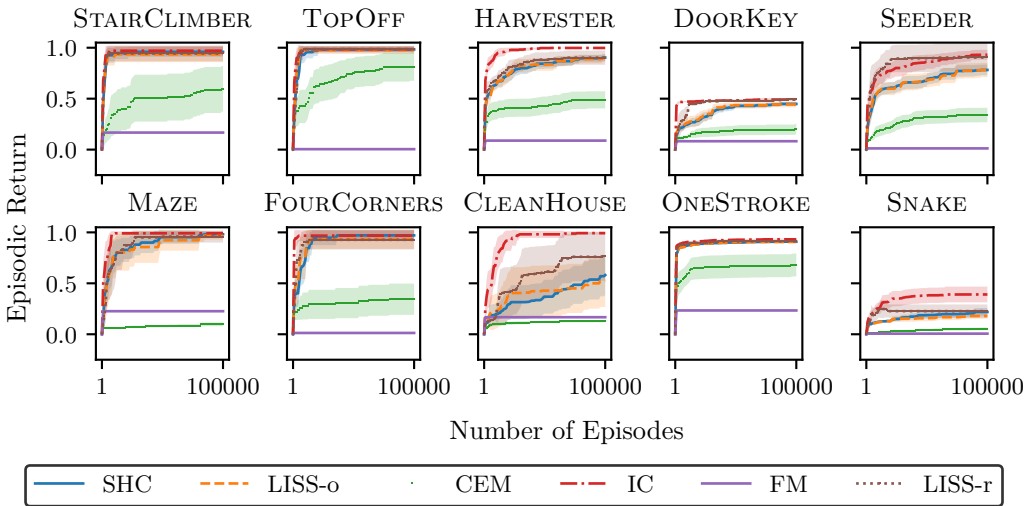

Figure 4: Average episodic return (maximum is 1.0 for all tasks) per number of episodes. The plots show average episodic return of 30 independent runs (seeds) and the 95% confidence interval.

INNATECODER is often much more sample-efficient than all baselines and, in many cases, by a large margin. We did not observe significant differences between IC-GPT and IC-Llama. LISS-o and LISS-r perform worse than INNATECODER and SHC in MicroRTS. However, LISS-o was competitive with SHC in Karel and LISS-r could outperform SHC (DoorKey and Seeder). This result suggests that the semantic space can be less conducive to search than the syntax space, depending on the quality of the options used to induce it. LISS-r performs better in Karel than in MicroRTS, probably because it uses a distribution $\mathcal{I}$ that uses a handcrafted probability distribution over the production rules of the language (Trivedi et al., 2021). The resulting grammar allows for the generation of helpful options. We do not have such a distribution for the MicroLanguage, which explains the results.

FM performs poorly in all experiments; the model is unable to generate effective policies in a zero-shot setting. The results of FM and INNATECODER support our hypothesis that INNATECODER can extract helpful options from foundation models even if the programs the model generates do not encode strong policies. In MicroRTS, some of the options allowed the agent to allocate units to collect resources and train other units. Other systems had to learn such skills from scratch, while INNATECODER's agent had them "innately available". The FM results also suggest that data contamination was not an issue in our experiments, as the model performed poorly on all tasks.

| INNATECODER | COAC | RAISocketAI | Mayari | Average |
|---|---|---|---|---|
| GPT-4o | 53.75 | 36.25 | 71.25 | 53.75 |
| Llama 3.1 | 43.79 | 70.00 | 58.17 | 57.32 |
| Llama 3.1 + GPT-4o | 70.14 | 72.92 | 46.39 | 63.15 |

Table 1: Winning rate of INNATECODER against winners of previous competitions, averaged across all 4 maps used in our experiments.

## 4.2 COMPETITION RESULTS

Table 1 shows the results of INNATECODER against the winners of previous MicroRTS competitions. The numbers are the average winning rate of INNATECODER against each system in the 4 maps used in our experiments. COAC and Mayari are human-written programmatic policies, and RAISocketAI is a DRL agent (Goodfriend, 2024). We used the RAISocketAI model submitted to the competition, which was trained with a larger computational budget than what we used with INNATECODER, thus giving RAISocketAI an advantage. We evaluated the models GPT-4o while generating 120 programs from which options are extracted, as well as Llama 3.1 while generating the same number of programs. Finally, we also evaluate INNATECODER when we take the union of the programs generated by both GPT-4o and Llama (denoted Llama 3.1 + GPT-4o in the table). The larger number of programs considered in the combination of GPT-4o and Llama 3.1 resulted in the best average winning rate.

The combination of programs written by Llama 3.1 and GPT-4o does not lead to "monotonic improvements", as evidenced by the drop in performance against Mayari. This happens because none of the competition winners is constrained by the DSL we use in our experiments. As a result, the optimization done in self-play might not be specific for the opponents evaluated in Table 1, but to policies written in the DSL and encountered during the self-play process.

## 4.3 EVALUATING NUMBER OF OPTIONS

Figure 5 presents the learning curves for different versions INNATECODER, where we vary the size of the set of options $\Omega$. The three lines with the highest winning rate are for option sets of sizes 5000, 7000, and 30000. The versions of INNATECODER with option sets of sizes 300, 600, and 1400 perform worse. These results demonstrate that INNATECODER can benefit from thousands of options. This is possible due to INNATECODER's way of using options through the induction of the language's underlying semantic space.

These results also show that INNATE-CODER's sample efficiency plateaus at 5000 options, since the use of 7000 and 30000 options does not increase performance. Interestingly, performance does not degrade either as we increase the set size. We conjecture that this occurs because, for large sets, many of the options will encode

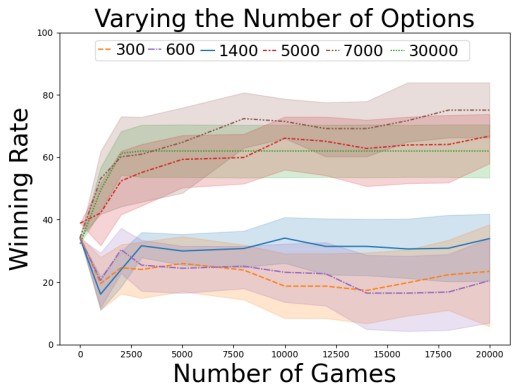

Figure 5: Average winning rate of INNATECODER policies for different sizes of the option set over 10 independent runs (seeds) of each version. We also present the 95% confidence intervals.

different and yet similar behaviors that do not affect the agent's winning rate. For example, an agent could use options $\omega_1$ or $\omega_2$ to achieve slightly different behaviors that lead to the same winning rate. That is, although the set of distinct behaviors encoded in the set options grows with larger sets, the relative number of options with behaviors that affect the winning rate remains roughly the same. As a result, the neighborhood function $\mathcal{N}_k^m$ that uses option sets of sizes 5000, 7000, or 30000 induces spaces that are similarly conducive to search algorithms. In Appendix F, we explain that the

difference in performance between INNATECODER with 1400 or fewer options and INNATECODER with 5000 or more is due to an increased chance of sampling helpful options with larger sets.

## 5  RELATED WORK

**Programmatic Policies**   One of the key challenges in generating programmatic policies is that the search space is discontinuous and gradient-based optimization cannot be used. Some previous work relied on imitation learning to guide the search for policies (Verma et al., 2018; 2019; Bastani et al., 2018; Milani et al., 2022; Liu et al., 2023d). The issue of this imitation learning approach is known as *representation gap* (Qiu & Zhu, 2022; Medeiros et al., 2022), where the space of programmatic policies does not include the oracle policy that the system tries to imitate. As a result, the oracle might guide the search to unpromising parts of the space. Previous work tried to learn latent spaces of programming languages that are conducive to search (Trivedi et al., 2021; Liu et al., 2023b), which was shown to be outperformed by the syntax space with SHC (Carvalho et al., 2024). Semantic spaces were shown to be more conducive to search than syntax spaces, but required a sequence of tasks, where the agent learns the space in one task and reuses it in others (Moraes & Lelis, 2024). Our work does not require an oracle agent nor a sequence of tasks to learn the semantic space.

**Options**   Options were shown to improve the sampling efficiency of learning agents through faster credit assignment (Mann & Mannor, 2014; Solway et al., 2014), better exploration (Baranes & Oudeyer, 2013; Bellemare et al., 2020), and transfer of knowledge across tasks (Konidaris & Barto, 2007; Alikhasi & Lelis, 2024). However, previous methods for learning options require the user to design them before learning starts (Sutton et al., 1999) or to provide considerable information as input to the process, such as the option duration (Frans et al., 2017; Tessler et al., 2017) or the number of options learned (Bacon et al., 2017; Igl et al., 2020). Other methods rely on the agent interaction with the current environment (Achiam et al., 2018; Machado et al., 2018; Jinnai et al., 2020) or with other earlier environments, as in transfer learning approaches (Konidaris & Barto, 2007; Alikhasi & Lelis, 2024). We present a novel way of learning options as they are not learned from the agent's experience nor designed by the user, but harnessed from foundation models. While we use options to define a search space, future work will explore their use as functions neural policies can call.

We provide additional related works on "foundation models as policies", "foundation models for planning", and "foundation models as search guidance" in Appendix A.

## 6  CONCLUSIONS

If given a single problem to solve, reinforcement learning agents start their learning process from scratch. They have to learn by interacting with the environment even the most basic skills to solve the problem. In this paper, we introduced INNATECODER, a system that equips learning agents with skills, in the form of programmatic options, before the agent starts to interact with the environment. This is achieved by extracting programmatic options from foundation models. We hypothesized that even if the model is unable to write programs encoding strong policies for a problem, sub-programs of the generated program could encode helpful options. We tested our hypothesis in MicroRTS and Karel, two domains in which programmatic policies represent the current state of the art. The policies INNATECODER generated outperformed, often by a large margin, a baseline that did not attempt to learn options; a baseline that learned the options while learning how to solve the problem; a baseline that learned the options from programs sampled directly from the domain-specific language; and the foundation model that attempted to generate programmatic policies directly. We also showed that some of the policies INNATECODER generated were competitive or outperformed the winners of previous MicroRTS competitions, including programmatic policies written by human programmers and a deep reinforcement learning agent that used a larger computational budget than we allowed INNATECODER to use. These results place INNATECODER as the current state-of-the-art in both Karel and MicroRTS. Our experiments also showed that INNATECODER's scheme of using programmatic options to induce semantic spaces allows it to benefit from thousands of options, while most previous work can benefit only from dozens or at most hundreds of options (Eysenbach et al., 2019).

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

## A ADDITIONAL RELATED WORK

**Foundation Models as Policies** Foundation models have been used to perceive, plan, and act (Park et al., 2023), often decomposing long-horizon goals into subtasks (Wang et al., 2023b), and/or integrating additional agent features such as memory (Zhu et al., 2023) and/or automatic learning curricula (Wang et al., 2023a). By contrast, INNATECODER uses the model as a pre-processing step to generate programmatic options, which make it more accessible due to the limited number of model calls. Foundation models have also been used to learn reward functions that are later used to

train agents (Klissarov et al., 2024). Similarly to our work, the model is used in a pre-processing step. However, in contrast with our work, it learns reward functions, while we learn programmatic options. Moreover, it needs a "diverse" set of states, which are generated by existing and proficient agents; INNATECODER learns in a zero-shot setting and only uses the generated options themselves to generate a set of states used to filter out the options encoding non-novel behaviors.

**Foundation Models for Planning**  Previous work has used foundation models for generating programmatic policies in the context of planning. For example, in generalized planning (GP), the goal is to synthesize programs that solve classical planning problems (Celorrio et al., 2019); foundation models have shown promise for GP (Silver et al., 2023). Foundation models has also been used successfully in the context of code generation for decision making in robotics (Liang et al., 2023; Singh et al., 2023). In contrast to these works, we do not attempt to use the model's generated program as a policy, but we extract options from them and use these options to induce a search space.

**Foundation Models as Search Guidance**  Foundation models have also been used to guide search algorithms. This includes methods for solving optimization problems (Yang et al., 2023; Guo et al., 2023) and to guide Monte Carlo tree search (Zhao et al., 2023). Foundation models were also used in genetic operators (Lehman et al., 2022; Liu et al., 2023c; Meyerson et al., 2023; Chen et al., 2023), including multi-objective (Liu et al., 2023a) and quality-diversity algorithms (Nasir et al., 2023). These works are resource-intensive due to calling the model during the search (Liu et al., 2023a). This contrasts with our work, which uses the model a small number of times in a pre-processing step.

# B  DEEP REINFORCEMENT LEARNING COMPARISON

## B.1  MICRORTS

To compare INNATECODER with a Deep Reinforcement Learning algorithm, we used the Gym-µRTS Huang et al. (2021). We evaluated INNATECODER with PPO Gridnet self-play using an encode-decode model. We chose this model because is the closest one to ours, as they both learn through self play. The DRL agents proposed by Huang et al. were specifically designed and tested for a map of size $16 \times 16$. We used the Gym-µRTS with the same settings presented in the repository, changing only the budget and the UnitTable used in the experiments. We trained both algorithms with a budget of 300 million steps in the MicroRTS.

We trained 15 DRL models for the BasesWorkers map ($16 \times 16$), using a Xeon 2.90GHz, 64GB of memory, and a dedicated Nvidia A10. Also, we performed 15 individual runs for INNATECODER. For each pair (DRL-INNATECODER) we ran 10 matches of the policies using the same evaluation used by Huang et al.. Each individual result is shown in Figure 6. Each run shows the number of losses, ties, and wins that INNATECODER achieved against DRL. The best score obtained by DRL is 5 losses and 5 wins presented in the individual run number 5 of graphs. In contrast to the policies INNATECODER generates, which can be used to play at any of the two locations of the map, the DRL-PPO agent is trained for a fixed position. This provides an advantage to DRL-PPO, as INNATECODER does not specialize in a given location of the map. Figure 7 shows the average results, where the whiskers show the 95% confidence interval. INNATECODER wins more than 70% of the matches with the PPO agent. Moreover, while the DRL agent needs around 3 days to train for 300 million steps, INNATECODER can perform the same number of training in less than 36 hours of computation using a single CPU.

## B.2  KAREL

For Karel the Robot, we compared INNATECODER with Hierarchical Programmatic Reinforcement Learning (HPRL) (Liu et al., 2023b), which uses neural and programmatic representations of policies. We evaluate the version of the system that uses PPO (HPRL-PPO), one of the best variants of HPRL. Table 2 shows the comparison between INNATECODER, HPRL-PPO, SHC (Carvalho et al., 2024), and CEM (Trivedi et al., 2021). These tests were performed with a budget of $10^5$ episodes. INNATECODER (IC in the table) obtained the highest average return in all tasks.

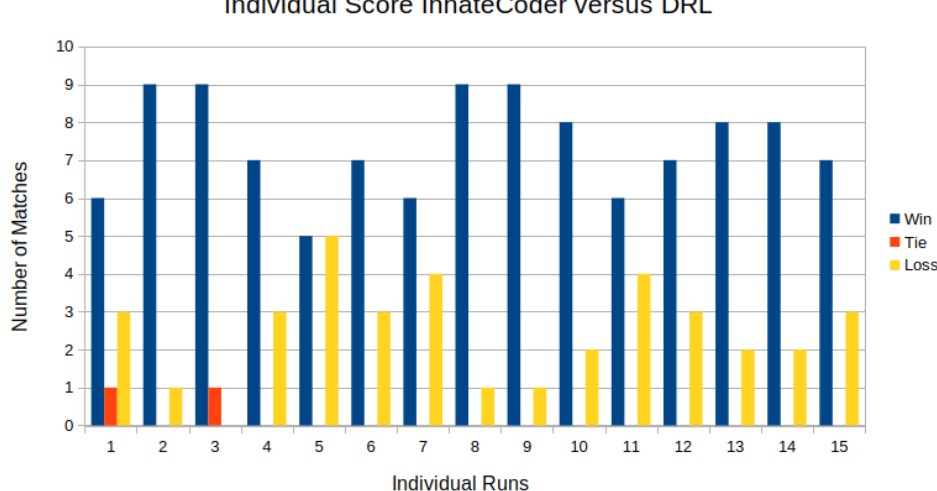

Figure 6: Individual results for INNATECODER against DRL in basesWorkers16x16A map. The columns are in order: win, tie, and loss, that INNATECODER got against DRL.

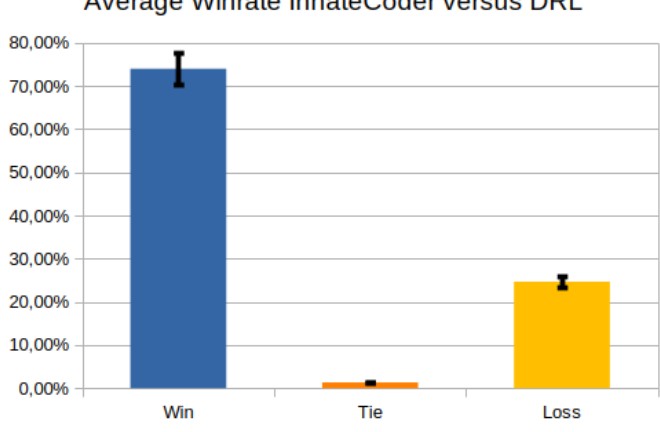

Figure 7: Average results of INNATECODER against DRL in basesWorkers16x16A map, with 95% confidence interval.

## C  CAN INNATECODER IMPROVE WITH MORE INFORMATION?

We evaluated whether INNATECODER's sample efficiency can scale with the amount of information we provide in the prompt used to generate the programmatic options. Figure 8 shows the results of two versions of INNATECODER: one that uses prompts with more information (IC+) and one that uses prompts with less information (IC-). The prompts are given in Section J.2 (more information) and Section J.3 (less information). The key difference between IC+ and IC- is that, in the former, we explain in the prompt how the agent can maximize its return. For example, in FourCorners we wrote "gent has to place one marker in each of the four corner cells of the grid". In contrast, in IC- we wrote "the robot will receive different reward values depending on its interactions inside the grid". Providing more information was never worse, and it was significantly better in two cases: Seeder and Snake. The ability to improve with more information is important because it allows the user of INNATECODER to achieve stronger results by crafting prompts that encode domain knowledge.

| Task | HPRL-PPO | SHC | CEM | IC |
|---|---|---|---|---|
| StairClimberSparse | **1.000** ±0.00 | **1.000** ±0.00 | 0.601 ±0.44 | **1.000** ±0.00 |
| MazeSparse | **1.000** ±0.00 | **1.000** ±0.00 | 0.097 ±0.03 | **1.000** ±0.00 |
| TopOff | **1.000** ±0.00 | **1.000** ±0.00 | 0.812 ±0.29 | **1.000** ±0.00 |
| FourCorners | **1.000** ±0.00 | **1.000** ±0.00 | 0.332 ±0.29 | **1.000** ±0.00 |
| Harvester | 0.924 ±0.13 | 0.906 ±0.07 | 0.487 ±0.17 | **1.000** ±0.00 |
| CleanHouse | 0.826 ±0.21 | 0.598 ±0.41 | 0.127 ±0.02 | **1.000** ±0.00 |
| DoorKey | 0.389 ±0.09 | 0.449 ±0.04 | 0.203 ±0.11 | **0.493** ±0.03 |
| OneStroke | 0.784 ±0.11 | 0.908 ±0.01 | 0.683 ±0.23 | **0.932** ±0.01 |
| Seeder | 0.539 ±0.17 | 0.779 ±0.10 | 0.339 ±0.14 | **0.931** ±0.09 |
| Snake | 0.283 ±0.18 | 0.217 ±0.08 | 0.053 ±0.02 | **0.391** ±0.15 |

Table 2: Mean and standard error of final episodic return of INNATECODER HPRL-PPO, SHC, and CEM in Karel problems.

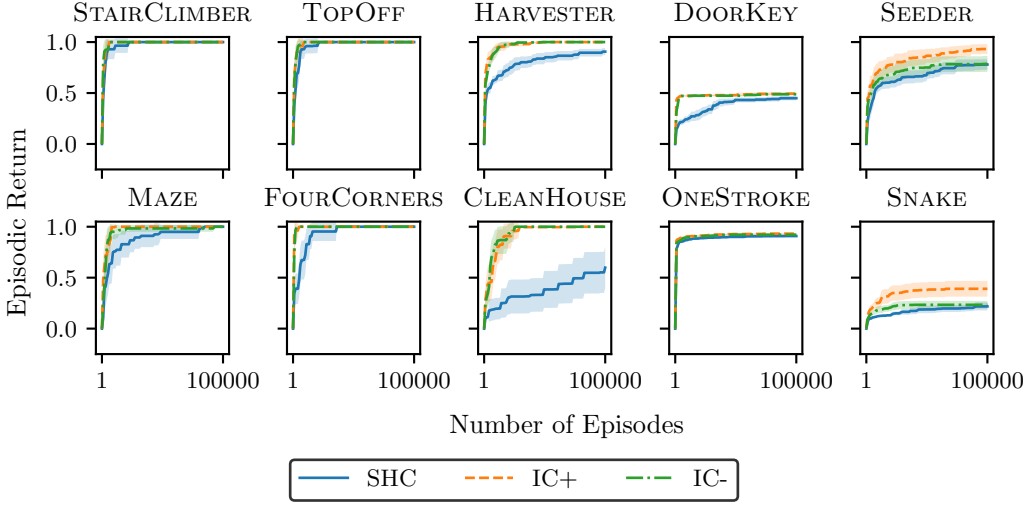

Figure 8: Evaluating INNATECODER with more (IC+) and less (IC-) information provided in the prompt used to harness programmatic options from the foundation model. We used GPT 4o in this experiment. Average episodic return (maximum is 1.0 for all tasks) per number of episodes. The plots show average episodic return of 30 independent runs (seeds) and the 95% confidence interval.

## D   CAN INNATECODER IMPROVE WITH MODEL SIZE?

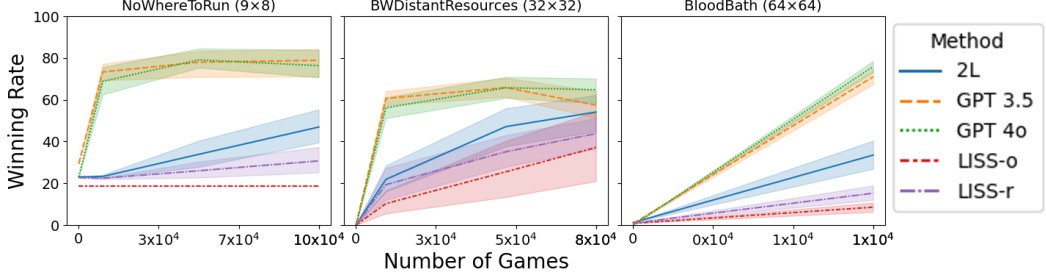

Figure 9:   The plots show the average winning rate of 30 independent runs (seeds) and the 95% confidence interval.

```
1  # InnateCoder's policy for the 64x64 map        1  # Llama's policy for the 64x64 map
2  for (Unit u)                                     2  for(Unit u)
3      u.attackIfInRange()                          3    for(Unit u)
4      for (Unit u)                                 4      u.train(Worker, Down, 5)
5          u.attackIfInRange()                      5    for(Unit u)
6          u.train(Heavy, EnemyDir, 50)             6      u.moveToUnit(Ally, Closest)
7      u.harvest(10)                                7      u.harvest(5)
8      if (u.OpponentHasUnitInPlayerRange())        8    for(Unit u)
9          pass                                     9      u.build(Barracks, Up, 1)
10     else                                         10     u.train(Ranged, EnemyDir, 10)
11         u.train(Worker, EnemyDir, 3)             11     u.attackIfInRange()
12         u.train(Ranged, EnemyDir, 15)            12   for(Unit u)
13     u.attack(Strongest)                          13     u.moveToUnit(Enemy, Weakest)
14     for (Unit u)                                 14     u.attack(Weakest)
15         u.harvest(5)
16     for (Unit u)
17         u.build(Barracks, EnemyDir, 8)
18     u.moveToUnit(Enemy, Strongest)
19     if (u.HasUnitInOpponentRange())
20         for (Unit u)
21             u.moveToUnit(Enemy, Farthest)
22             u.train(Light, EnemyDir, 6)
23     u.train(Worker, Down, 6)
```

Figure 10: Left: A programmatic policy INNATECODER generated for the largest 64×64 map. This policy defeats the last three winners of the MicroRTS Competition: COAC, Mayari, and RAISocketAI. Right: one of the policies Llama 3.1 generated for the same map.

We evaluated GPT 3.5-turbo and GPT 4o on the MicroRTS tasks. Figure 9 shows the results, where we report the average winning rate and the 95% confidence intervals of 30 independent runs (seeds). Interestingly, we do not notice a significant change in winning rate as we move from the larger GPT 4o to the smaller GPT 3.5.

# E  SAMPLES OF PROGRAMS

Figure 10 shows an example of a programmatic policy INNATECODER generated for the BloodBath 64×64 map (left), and a program Llama 3.1 generated for the same map (right). The policy INNATECODER generated defeats the last three winners of the MicroRTS competition: COAC, Mayari, and RAISocketAI. This policy presents non-trivial features. For example, lines 7-11 will train Worker and Ranged units only if the player is not engaged in combat. This means that this policy focuses on economy and on Ranged units in the early stages of the game. Later in the game, the agent will save its resources to train Light units (line 21). Light units can be trained and move more quickly than Ranged units. While a Light unit is trained in 80 time steps of the game, a Ranged unit requires 100 time steps to be trained; also, a Light unit can move one cell every 8 time steps, while Ranged units move one cell every 10 time steps. Light units allow for a faster return of the resources invested.

The program the foundation model generates represents a weak policy (program shown on the right of Figure 10). However, even weak policies can contain pieces of code—options—that can be helpful while searching for strong policies to play the game. For example, lines 2 and 3 offer a prioritization scheme for investing resources to train Worker units because it will iterate over all units until it finds the Base, which will be used to train up to 5 workers. Lines 2 and 3 are often found in strong policies. For example, the program shown on the left of Figure 10 shows a similar structure in lines 1 and 10.

# F  OPTION USAGE IN PROGRAMMATIC POLICIES

Figure 11 shows an example of a program INNATECODER generated as the final policy (the one that the system produces as output) for the LetMeOut map (16 × 8), where four options are used. The colored lines represent the options. Lines 3 and 4 form one option (blue), while lines 8 and 9 (red), 11 and 12 (purple), and 13 and 14 (green) form the other three options. In this representative example, 8 of 14 lines are options, which represents 57% of the lines. We also analyzed 10 independent runs of

```
1   for(Unit u)
2     u.harvest(1)
3     if(HasNumberOfUnits(Barracks,1))
4       u.train(Ranged,Down,10)
5   for(Unit u)
6     for(Unit u)
7       u.moveToUnit(Ally,MostHealthy)
8       for(Unit u)
9         u.attack(Closest)
10      for(Unit u)
11        for(Unit u)
12          u.train(Worker,EnemyDir,3)
13        for(Unit u)
14          u.moveToUnit(Ally,Weakest)
```

Figure 11: Programmatic policy INNATECODER generated for the LetMeOut map ($16 \times 8$). The colored lines represent options from one execution of INNATECODER that used 5000 options.

INNATECODER in the $9 \times 8$ map with an initial option set extracted from 120 policies, which were generated with the Llama 3.1 model. We found that on average, 63% of the options in the initial set are used in a best response during the self-play training process. Recall that a best response is the solution INNATECODER finds to a given MDP within the self-play algorithm. The maximum percentage of options used in the 10 runs was 73% and the minimum was 46%. Even if an option is not used in the policy INNATECODER outputs, the option can still have played an important role in allowing INNATECODER arrive at the output it produced. This is because an option could have been part of one of the best responses encountered during self play, and these best responses provide the signal needed to guide the search toward stronger policies for playing the game (Moraes et al., 2023).

The policy shown in Figure 11 also provides an explanation for the difference in performance between the versions of INNATECODER that are initialized with a pool of 1400 or fewer options and the versions that are initialized with a pool of 5000 or more options (see Figure 5). The options highlighted in Figure 11 are in the set of 5000 options of an INNATECODER run, but they are not in the set of 1400 options of another run of the algorithm. These options are clearly helpful because they appear in the output policy. Sampling 5000 options instead of 1400 increases the chances of adding some of these helpful options to the library, thus explaining the gap we see between INNATECODER with 1400 options or fewer and INNATECODER with 5000 options or more in Figure 5.

## G  VARYING THE VALUE OF $\epsilon$

Figure 12 shows the performance of INNATECODER as we vary the value of $\epsilon$. Recall that $\epsilon$ dictates how much of the search is done in the syntax space and how much of the search is done in the semantic space. Also, recall that smaller values of $\epsilon$ mean that the semantic space is used more often. There are two groups of lines at 80000 games played: $\epsilon$-values of 0.3, 0.4, 0.5, and 0.6 (top lines) and 0.1, 0.2, and 0.7 (bottom lines). These results suggest that INNATECODER is robust to the choice of $\epsilon$ value, as a wide range of values produce competitive results among themselves. The results also suggest that if the value of $\epsilon$ is too small, then the search does not sufficiently explore the syntax space to eventually find programs that were not originally in the library of options. If the value of $\epsilon$ is too large, then the search is not making use of the helpful options in the library as often as it could.

## H  DOMAIN-SPECIFIC LANGUAGES (DSLs)

In this section, we present the DSLs used in our experiments for both Karel the Robot and MicroRTS.

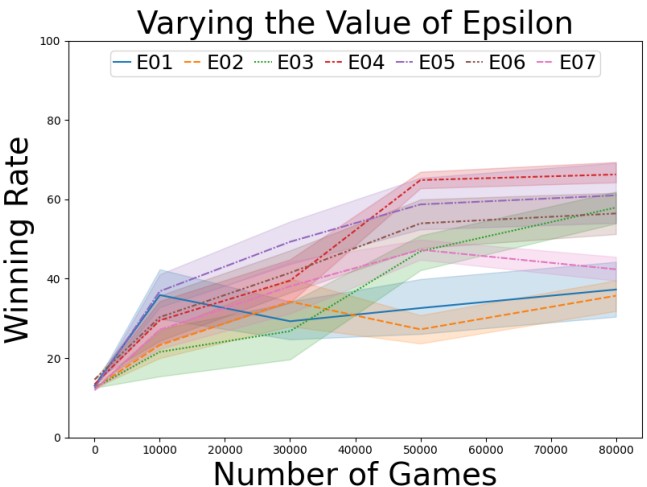

Figure 12: INNATECODER with values of $\epsilon$ in $\{0.1, 0.2, 0.3, 0.4, 0.5, 0.6, 0.7\}$ in the $9{\times}8$ MicroRTS map. Here, each INNATECODER line is evaluated as in the "Efficiency Experiment" (Figure 3). Each line shows the average of 30 independent runs and the 95% confidence intervals.

### H.1   KAREL

The context-free grammar below presents the DSL for Karel the Robot. This DSL is the same as that used in the previous work (Trivedi et al., 2021; Liu et al., 2023b).

$$
\begin{aligned}
\text{Program } \rho &:= \texttt{DEF run m( } s \texttt{ m)} \\
\text{Statement } s &:= \texttt{WHILE c( } b \texttt{ c) w( } s \texttt{ w) } \mid \texttt{IF c( } b \texttt{ c) i( } s \texttt{ i) } \mid \\
&\qquad \texttt{IFELSE c( } b \texttt{ c) i( } s \texttt{ i) ELSE e( } s \texttt{ e) } \mid \texttt{REPEAT R=}n \texttt{ r( } s \texttt{ r) } \mid \\
&\qquad s; s \mid a \\
\text{Condition } b &:= h \mid \texttt{not( } h \texttt{ )} \\
\text{Number } n &:= 1 \mid 2 \mid 3 \mid \ldots \mid \texttt{infinity} \\
\text{Perception } h &:= \texttt{frontIsClear} \mid \texttt{leftIsClear} \mid \texttt{rightIsClear} \mid \\
&\qquad \texttt{markersPresent} \mid \texttt{noMarkersPresent} \\
\text{Action } a &:= \texttt{move} \mid \texttt{turnLeft} \mid \texttt{turnRight} \mid \texttt{putMarker} \mid \texttt{pickMarker}
\end{aligned}
$$

### H.2   MICRORTS

The context-free grammar below presents the DSL for MicroRTS. This DSL is the same as that used in recent work (Moraes & Lelis, 2024). Note that an early version of the Microlanguage appears in the work of Mariño et al. (2021), which was published prior to the cut-off date of the ChatGPT model used in our experiments. However, this earlier version of the language is fundamentally different from the one used in recent work and in our experiments. For context, the interpreter we use in our experiments cannot run the programs written in the language used of Mariño et al. (2021). To illustrate some differences, the instruction `moveToUnit(Light, Ally, strongest, u)` in the older Microlanguage has four parameters, while in ours it has only two. The older language has a larger collection of high-level functions, such as `HaveQtdUnitsbyType`, which our version of the language does not have. Considering these key differences, it is unlikely that the data from Mariño et al. (2021) have influenced our results more than the vast collection of programs written in various programming languages that are present in the corpus used to train these foundation models.

$$S \to SS \,|\, \texttt{for(Unit u) S} \,|\, \texttt{if(B) then S}$$
$$\,|\, \texttt{if(B) then S else S} \,|\, C \,|\, \lambda$$
$$B \to \texttt{hasNumberOfUnits}(T, N) \,|\, \texttt{opponentHasNumberOfUnits}(T, N)$$
$$\,|\, \texttt{hasLessNumberOfUnits}(T, N) \,|\, \texttt{haveQtdUnitsAttacking}(N)$$
$$\,|\, \texttt{hasUnitWithinDistanceFromOpponent}(N)$$
$$\,|\, \texttt{hasNumberOfWorkersHarvesting}(N)$$
$$\,|\, \texttt{is\_Type}(T) \,|\, \texttt{isBuilder()}$$
$$\,|\, \texttt{canAttack()} \,|\, \texttt{hasUnitThatKillsInOneAttack()}$$
$$\,|\, \texttt{opponentHasUnitThatKillsUnitInOneAttack()}$$
$$\,|\, \texttt{hasUnitInOpponentRange()}$$
$$\,|\, \texttt{opponentHasUnitInPlayerRange()}$$
$$\,|\, \texttt{canHarvest()}$$
$$C \to \texttt{build}(T, D, N) \,|\, \texttt{train}(T, D, N) \,|\, \texttt{moveToUnit}(T_p, O_p) \,|\, \texttt{attack}(O_p)$$
$$\,|\, \texttt{harvest}(N) \,|\, \texttt{attackIfInRange()} \,|\, \texttt{moveAway()}$$
$$T \to \texttt{Base} \,|\, \texttt{Barracks} \,|\, \texttt{Ranged} \,|\, \texttt{Heavy}$$
$$\,|\, \texttt{Light} \,|\, \texttt{Worker}$$
$$N \to 0 \,|\, 1 \,|\, 2 \,|\, 3 \,|\, 4 \,|\, 5 \,|\, 6 \,|\, 7 \,|\, 8 \,|\, 9$$
$$\,|\, 10 \,|\, 15 \,|\, 20 \,|\, 25 \,|\, 50 \,|\, 100$$
$$D \to \texttt{EnemyDir} \,|\, \texttt{Up} \,|\, \texttt{Down} \,|\, \texttt{Right} \,|\, \texttt{Left}$$
$$O_p \to \texttt{Strongest} \,|\, \texttt{Weakest} \,|\, \texttt{Closest} \,|\, \texttt{Farthest}$$
$$\,|\, \texttt{LessHealthy} \,|\, \texttt{MostHealthy} \,|\, \texttt{Random}$$
$$T_p \to \texttt{Ally} \,|\, \texttt{Enemy}$$

# I    KAREL PROBLEM SETS

The KAREL problem sets (Trivedi et al., 2021; Liu et al., 2023b) are divided into two parts— KAREL and KAREL-HARD. KAREL consists of six different tasks, while KAREL-HARD includes four additional tasks that are comparatively more difficult to solve. In this section, we describe the initial state and the return function of each task in both KAREL and KAREL-HARD problem sets.

## I.1    KAREL

**StairClimber.**    In this task, the agent operates within a $12 \times 12$ grid containing stairs formed by walls. The goal for the agent is to reach a marker located above its position on the stairs. The initial positions of the agent and the marker are randomly initialized on the stairs. The agent receives an episodic return of $1$ if it successfully reaches the marker, and $0$ otherwise. Moving to a position outside the contour of the stairs results in a return of $-1$.

**FourCorners.**    In this task, the goal for the agent is to place a marker in the four corners of a $12 \times 12$ grid. The initial position of the agent is randomly initialized near the wall. The return is calculated as the number of corners with one marker divided by four.

**TopOff.**    In this task, the agent is always initialized on the bottom left of a $12 \times 12$ grid, and the markers are placed randomly on the bottom row of the grid. The goal for the agent to place markers on top of the markers present in that row. The return is calculated as the number of markers topped off divided by the number of markers present in the grid.

**Maze.**    A maze, formed by walls, is randomly configured on a $12 \times 12$ grid, and a marker and an agent are randomly placed within it. The goal for the agent is to reach the marker, that yields an episodic return of $1$. Otherwise, the episodic return is $0$.

**CleanHouse.**    Markers and walls are randomly placed in a $22 \times 14$ grid, referred to as the apartment. The position of the agent is also initialized randomly. Its goal is to pick all the markers inside the apartment. The return is calculated as the number of picked markers divided by the total number of markers present.

**Harvester.**    In this task, the agent operates within an $8 \times 8$ grid filled with markers. The agent is placed randomly on the bottom row, with the goal of collecting all markers in the grid. The return is calculated as the number of collected markers divided by the total number of cells in the grid.

## I.2    KAREL-HARD

**DoorKey.**    The environment, consisting of an $8 \times 8$ grid is divided into two chambers by a vertical bar with a door. The position of the agent is initialized randomly, and a marker is placed randomly in each chamber. The goal for the agent is to pick the marker in the left chamber, which will open the door in the vertical bar, and then pick the marker in the right chamber. Picking each marker results in a return of $0.5$.

**OneStroke.**    In this task, the goal for the agent is to visit every cell of an $8 \times 8$ grid without repetition. Once a cell is visited, it transforms into a wall, and the episode terminates if the agent hits a wall. The return is calculated as the number of visited cells divided by the total number of cells in the grid.

**Seeder.**    The environment is given by an $8 \times 8$ empty grid with an agent initialized at a random position. The goal for the agent is to place a marker in each cell of the grid. The return is calculated as the number of placed markers divided by the total number of cells.

**Snake.**    In this task, an $8 \times 8$ grid is initialized with an agent and a marker at random positions. The agent behaves like the head of a snake, with its body growing after collecting a marker. Once a marker is collected, another marker is placed at a random position. This process continues until the agent collects 20 markers. The goal for the agent is to collect the markers without hitting its own body. The return is calculated as the number of collected markers divided by 20.

## J    KAREL PROMPTS

In this section, we present the prompts used to generate a program that encodes a policy. These are the prompts we used multiple times to obtain a set of programmatic policies. This section is divided into three subsections. Section J.1 includes the complete prompt used to obtain a program from the model to solve the 'Seeder' task. For each task, we provide two types of prompts: one with more information and another with less information. Section J.2 contains the details of the environment with more information for each task, while Section J.3 contains the prompts with less information for each task.

### J.1    COMPLETE PROMPT FOR THE SEEDER TASK

The prompt for obtaining a program that encodes a policy for the 'Seeder' task is shown below. Note that the first paragraph of the prompt explains the environment. Only this paragraph is changed from one task to the next.

> Consider an 8x8 gridded 2D environment in Karel the Robot, where an agent has to place one marker in cell of the grid. The grid is initially empty and the initial position of the agent is randomly assigned at the beginning of each episode.

The following is the Context Free Grammar (CFG) for the Karel domain:

$P \rightarrow$ run $\{S\}$

$S \rightarrow$ WHILE (B) $\{S\}$ | S S | A | REPEAT R=N $\{S\}$ | IF (B) $\{S\}$ | IF (B) $\{S\}$ ELSE $\{S\}$

$B \rightarrow$ H | not H

$H \rightarrow$ frontIsClear | leftIsClear | rightIsClear | markersPresent | noMarkersPresent

$A \rightarrow$ move | turnLeft | turnRight | putMarker | pickMarker

$N \rightarrow 1 \mid 2 \mid 3 \mid ... \mid$ infinity

The CFG is explained below in the "CFG Explanation" section:

CFG Explanation:
P: The main program named as "run" which contains Statement S
S: Consists of statements such as WHILE, IF, IFELSE, REPEAT. Can have multiple statements "S S" or action "A"
B: Perception H or not perception H that returns true or false
H: Some boolean variables that provide the idea of the environment with true or false
A: Action to be taken by the agent
N: A positive integer that indicates the number of repetitions
frontIsClear: Checks if the next cell towards the direction the agent is facing is inside the grid
leftIsClear: Checks if the next cell to the left of the direction the agent is facing is inside the grid
rightIsClear: Checks if the next cell to the right of the direction the agent is facing is inside the grid
move: The agent moves towards its front
turnLeft: The agent turns left
turnRight: The agent turns right
putMarker: The agent puts a marker in the current cell
pickMarker: The agent picks a marker from the current cell
...: It is not part of the CFG. It has been used to indicate all positive numbers in between.

To write a program from the given CFG, the following "Program Writing Guidelines" must be followed:

Program Writing Guidelines:
1. In the CFG, 'infinity' means that the value of N can be up to infinity. So do not write any part of the program like "R=infinity"
2. DO NOT write "..." in the program, since it is not a part of the CFG
3. The program must start with "run"

Now your tasks are the following 3:

1. Read carefully about the details of the environment, the CFG and its explanation.
2. Follow the CFG and the program writing guidelines and write a program of maximum 8 lines that will gain the maximum reward inside this given environment.
3. Write the program inside ¡program¿¡/program¿ tag.

## J.2 ENVIRONMENT DETAILS WITH MORE INFORMATION

### STAIRCLIMBER

Consider a 12x12 gridded 2D environment in Karel the Robot, where an agent has to reach a marker by climbing up along a stair. The grid contains a stair-like structure and the initial position of the agent is randomly assigned near the stair with the marker placed at the higher end, at the beginning of each episode.

### FOURCORNERS

Consider a 12x12 gridded 2D environment in Karel the Robot, where an agent has to place one marker in each of the four corner cells of the grid. The grid is initially empty and the initial position of the agent is randomly assigned near the wall of the grid at the beginning of each episode.

### TOPOFF

Consider a 12x12 gridded 2D environment in Karel the Robot, where an agent has to place markers on top of other cells that already have markers, and then reach at the rightmost cell of the bottom row. The markers are initialized randomly at the bottom row. The position of the agent is fixed at the leftmost cell of the bottom row at the beginning of each episode.

### MAZE

Consider a 12x12 gridded 2D environment in Karel the Robot, where an agent has to pick a marker following a path surrounded by walls. Some cells of the grid are filled with walls, collectively referred to as the maze. The agent, the marker and the maze are randomly placed at the beginning of each episode.

### CLEANHOUSE

Consider a 22x14 gridded 2D environment in Karel the Robot, where an agent has to pick some markers that are placed randomly inide the grid. There are also some cells filled with obstacles. The position of the agent is fixed whereas, the markers are initialized at random cells at the beginning of each episode.

### HARVESTER

Consider an 8x8 gridded 2D environment in Karel the Robot, where an agent has to pick one marker from every cell of the grid. The grid is initially filled with markers and the initial position of the agent is randomly assigned at the beginning of each episode.

### DOORKEY

Consider an 8x8 gridded 2D environment divided into two chambers by walls in Karel the Robot, where an agent has to pick a marker from the left chamber and put it over another marker placed at the right chamber. The agent cannot access the right chamber without picking the marker from the left chamber. The position of the agent, the marker of the left

chamber and the marker of the right chamber are initialized randomly at the beginning of each episode.

### ONESTROKE

Consider an 8x8 gridded 2D environment in Karel the Robot, where an agent has to visit as many cells as possible in one attempt. Once the agent visits a cell, it will be filled with a wall. The position of the agent is initialized randomly at the beginning of each episode.

### SEEDER

Consider an 8x8 gridded 2D environment in Karel the Robot, where an agent has to place one marker in every cell of the grid. The grid is initially empty and the initial position of the agent is randomly assigned at the beginning of each episode.

### SNAKE

Consider an 8x8 gridded 2D environment in Karel the Robot, where an agent has to pick a marker multiple times inside the grid. The position of the marker will be changed once the agent picks the marker. Each time the agent picks a marker, it will be attached to its body. For instance, if the agent picks 3 markers, it will have 3 markers added to its back. The agent has to pick as many markers as possible without hitting the markers attached to its body. The position of the agent and the marker are initialized randomly at the beginning of each episode.

### J.3 ENVIRONMENT DETAILS WITH LESS INFORMATION

### STAIRCLIMBER

Consider a 2D 12x12 grid with an agent placed randomly in any cell, which will interact within the grid. Each cell may contain a marker, be empty, or be blocked by a wall. Initially, for this particular problem, there is a stair-like structure and the agent is placed near the stair. The robot will receive different reward values depending on its interactions inside the grid. The interaction of the robot will be decided by a program written in the domain-specific language to be provided below as a context-free grammar (CFG). The goal is to generate a program that will maximize the sum of rewards the robot obtains by following that program.

### FOURCORNERS

Consider a 2D 12x12 grid with an agent placed randomly in any cell, which will interact within the grid. Each cell may contain a marker, be empty, or be blocked by a wall. Initially, for this particular problem, no markers are present in any cell of the grid.
The robot will receive different reward values depending on its interactions inside the grid. The interaction of the robot will be decided by a program written in the domain-specific language to be provided below as a context-free grammar (CFG). The goal is to generate a program that will maximize the sum of rewards the robot obtains by following that program.

TOPOFF

> Consider a 2D 12x12 grid with an agent placed at a fixed cell at the leftmost cell of the bottom row, which will interact within the grid. Each cell may contain a marker, be empty, or be blocked by a wall. Initially, for this particular problem, there are markers present in some cells at the bottom row of the grid.
>
> The robot will receive different reward values depending on its interactions inside the grid. The interaction of the robot will be decided by a program written in the domain-specific language to be provided below as a context-free grammar (CFG). The goal is to generate a program that will maximize the sum of rewards the robot obtains by following that program.

MAZE

> Consider a 2D 12x12 grid with an agent placed randomly in any cell, which will interact within the grid. Each cell may contain a marker, be empty, or be blocked by a wall. Initially, for this particular problem, there are few walls and a marker inside the grid.
>
> The robot will receive different reward values depending on its interactions inside the grid. The interaction of the robot will be decided by a program written in the domain-specific language to be provided below as a context-free grammar (CFG). The goal is to generate a program that will maximize the sum of rewards the robot obtains by following that program.

CLEANHOUSE

> Consider a 2D 22x14 grid with an agent placed at a fixed cell, which will interact within the grid. Each cell may contain a marker, be empty, or be blocked by a wall. Initially, for this particular problem, there are some markers and obstacles randomly placed inside the grid.
>
> The robot will receive different reward values depending on its interactions inside the grid. The interaction of the robot will be decided by a program written in the domain-specific language to be provided below as a context-free grammar (CFG). The goal is to generate a program that will maximize the sum of rewards the robot obtains by following that program.

HARVESTER

> Consider a 2D 8x8 grid with an agent placed randomly in any cell, which will interact within the grid. Each cell may contain a marker, be empty, or be blocked by a wall. Initially, for this particular problem, a marker is present in each cell of the grid.
>
> The robot will receive different reward values depending on its interactions inside the grid. The interaction of the robot will be decided by a program written in the domain-specific language to be provided below as a context-free grammar (CFG). The goal is to generate a program that will maximize the sum of rewards the robot obtains by following that program.

DOORKEY

> Consider a 2D 8x8 grid divided into two chambers with an agent placed randomly in the left chamber, which will interact within the grid. Each cell may contain a marker, be empty, or be blocked by a wall. Initially, for this particular problem, there is one marker in each chamber of the grid.
>
> The robot will receive different reward values depending on its interactions inside the grid. The interaction of the robot will be decided by a program written in the domain-specific language to be provided below as a context-free grammar (CFG). The goal is to generate a program that will maximize the sum of rewards the robot obtains by following that program.

ONESTROKE

> Consider a 2D 8x8 grid with an agent placed randomly in any cell, which will interact within the grid. Each cell may contain a marker, be empty, or be blocked by a wall. Initially, for this particular problem, no markers are present in any cell of the grid. Cells that will be visited by the agent will be filled with obstacles.
>
> The robot will receive different reward values depending on its interactions inside the grid. The interaction of the robot will be decided by a program written in the domain-specific language to be provided below as a context-free grammar (CFG). The goal is to generate a program that will maximize the sum of rewards the robot obtains by following that program.

SEEDER

> Consider a 2D 8x8 grid with an agent placed randomly in any cell, which will interact within the grid. Each cell may contain a marker, be empty, or be blocked by a wall. Initially, for this particular problem, no markers are present in any cell of the grid.
>
> The robot will receive different reward values depending on its interactions inside the grid. The interaction of the robot will be decided by a program written in the domain-specific language to be provided below as a context-free grammar (CFG). The goal is to generate a program that will maximize the sum of rewards the robot obtains by following that program.

SNAKE

> Consider a 2D 8x8 grid with an agent placed randomly in any cell, which will interact within the grid. Each cell may contain a marker, be empty, or be blocked by a wall. Initially, for this particular problem, there is one marker inside the grid. The position of the marker changes depending on a certain behaviour of the agent.
>
> The robot will receive different reward values depending on its interactions inside the grid. The interaction of the robot will be decided by a program written in the domain-specific language to be provided below as a context-free grammar (CFG). The goal is to generate a program that will maximize the sum of rewards the robot obtains by following that program.

## K MICRORTS MAPS

Figure 13 shows the three MicroRTS maps used in our experiments. In these maps, the geometric shapes with blue borders represent the units of one player, while those bounded by red borders represent the units of the other player. Neutral objects do not have colored borders (e.g., resources in light green and walls in dark green).

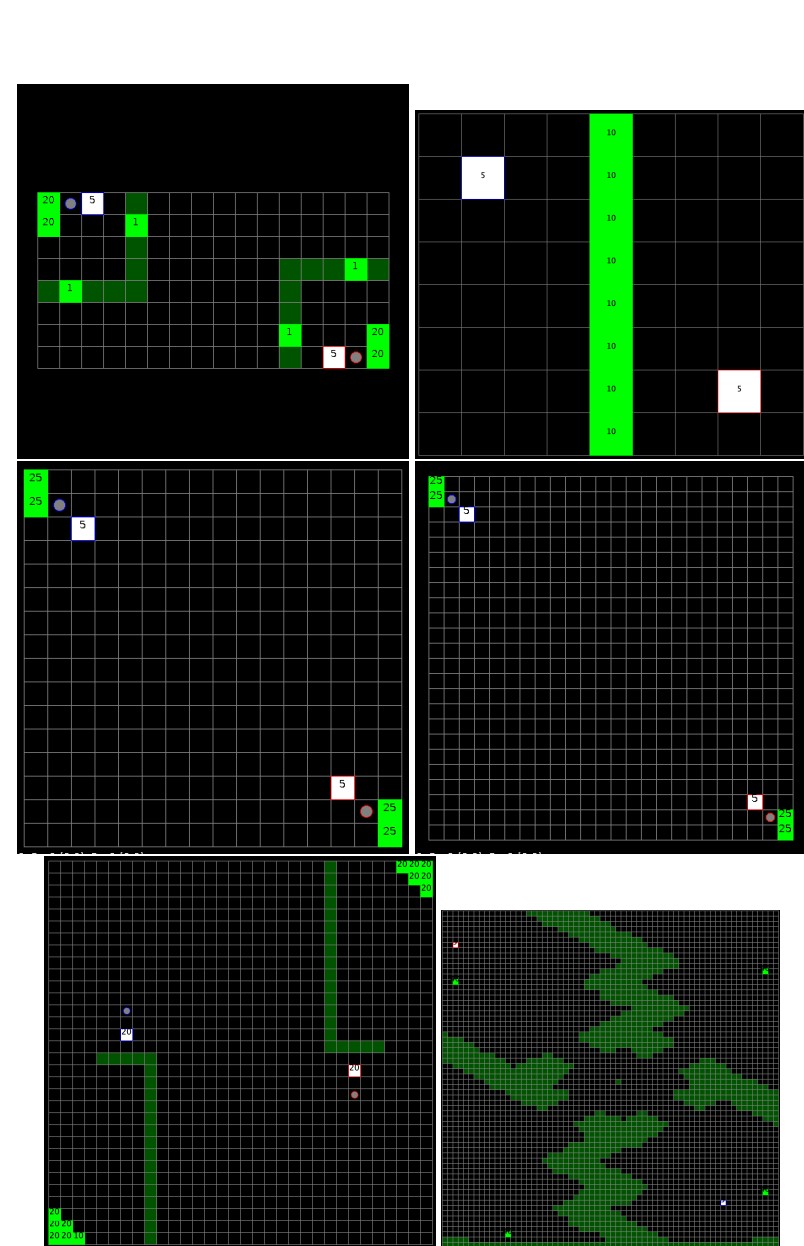

Figure 13: From left to right: LetMeOut (16×8), NoWhereToRun (9×8), BasesWorkers(16×16), BasesWorkers(24×24), BWDistantResources (32×32), BloodBath (64×64)

## L   MICRORTS PROMPTS

In this section, we present the prompt used to obtain one program that encodes a policy for MI-CRORTS. This is the prompt that we use multiple times to obtain a set of programmatic policies. Section L.1 shows the complete prompt for the 'NoWhereToRun (9x8)' map, as a complete example. Section L.2 shows the description of the environment used in the prompts of each map.

### L.1   COMPLETE PROMPT FOR THE NOWHERETORUN (9X8) MAP

The prompt for obtaining a program that encodes policies for the 'NoWhereToRun (9x8)' map is shown below. Note that the first paragraph of the prompt mentions the details of the environment for a given map. Only this paragraph describing the environment is updated to obtain the program for each separate map. We use a subset of the MicroLanguage, where conditionals are removed. Our goal was to have a language that would be easier for the foundation model to use.

---

Consider a 9x8 gridded map of microRTS, a real-time strategy game. Consider this map as a 2-dimensional array with the following structure:

- There are a total of 8 neutral resource cells situated along the central column of the map, dividing the map into two parts. Each resource cell contains 10 units of resources.

- The base B1 of player 1 is located at index (1,1), which is located on the left side of the map.

- The base B2 of player 2 is located at index (7,6), which is located on the right side of the map.

- Each player controls one base, which initially has 5 units of resources.

- The only unit a player controls at the beginning of the game is the base.

Consider this Context-Free Grammar (CFG) describing a programming language for writing programs encoding strategies of microRTS. The CFG is shown in the $< CFG >< /CFG >$ tag bellow:

$< CFG >$
$$S \rightarrow SS \mid \text{for(Unit u) S} \mid C \mid \lambda$$
$$C \rightarrow u.\text{build}(T, D, N) \mid u.\text{train}(T, D, N) \mid u.\text{moveToUnit}(T_p, O_p)$$
$$\mid u.\text{attack}(O_p) \mid u.\text{harvest}(N) \mid u.\text{attackIfInRange}() \mid u.\text{moveAway}()$$
$$T \rightarrow \text{Base} \mid \text{Barracks} \mid \text{Ranged} \mid \text{Heavy}$$
$$\mid \text{Light} \mid \text{Worker}$$
$$N \rightarrow 0 \mid 1 \mid 2 \mid 3 \mid 4 \mid 5 \mid 6 \mid 7 \mid 8 \mid 9$$
$$\mid 10 \mid 15 \mid 20 \mid 25 \mid 50 \mid 100$$
$$D \rightarrow \text{EnemyDir} \mid \text{Up} \mid \text{Down} \mid \text{Right} \mid \text{Left}$$
$$O_p \rightarrow \text{Strongest} \mid \text{Weakest} \mid \text{Closest} \mid \text{Farthest}$$
$$\mid \text{LessHealthy} \mid \text{MostHealthy} \mid \text{Random}$$
$$T_p \rightarrow \text{Ally} \mid \text{Enemy}$$
$< /CFG >$

This language allows nested loops. It contains several command-oriented functions (C). The Command functions ('C' in the CFG) are described below:

1. u.build(T, D, N): Builds N units of type T on a cell located in the D direction of the unit. The u.build function is used to build Barracks and Base.

2. u.train(T, D, N): Trains N units of type T on a cell located in the D direction of the structure responsible for training them. For example, the instruction u.train(Heavy, Down, 1) will allow the agent to train at most 1 heavy unit in the down direction of

---

the Barrack, while the instruction u.train(Heavy, EnemyDir, 20) will allow to train at most 20 towards the direction of the opponent. The number used in the function calls could play a big role in the strategy the program encodes. The u.train function is used to train Worker, Ranged, Light, and Heavy units.

3. u.moveToUnit(T_p, O_p): Commands a unit to move towards the player T_p following a criterion O_p.

4. u.attack(O_p): Sends N Worker units to harvest resources.

5. u.harvest(N): Sends N Worker units to harvest resources. For example, u.harvest(5) will send 5 workers to harvest resources.

6. u.attackIfInRange(): Commands a unit to stay idle and attack if an opponent unit comes within its attack range.

7. u.moveAway(): Commands a unit to move in the opposite direction of the player's base.

'T' represents the types a unit can assume.
'D' represents the directions available used in action functions.
'O_p' is a set of criteria to select an opponent unit based on their current state.
'T_p' represents the set of target players.
'N' is the number of units that can be any integers from 0 to 10, or 15, or 20, or 25, or 50, or 100.

The following 4 are some guidelines for writing the playing strategy:

1. There is NO NEED TO write classes, or initiate objects such as Unit, Worker, etc. There is also NO NEED TO write comments.

2. Use curly braces like C/C++/Java while writing any 'for' block. Start the curly braces in the same line of the block.

3. A strategy must be written inside one or multiple 'for' blocks.

4. This language does not have if-statements.

Now your tasks are the following 8:

1. Understand the command (C) functions from above and try to relate them in the context of playing strategies for a real-time strategy game.

2. Write a program in the microRTS language encoding a very strong game-playing strategy for the map described above. You must follow the guidelines of writing the playing strategy while writing your program.

3. You must not use any symbols (for example &&, ||, etc.) that the CFG does not accept. You have to strictly follow the CFG while writing the program.

4. Look carefully, the methods of non-terminal symbols C have prefixes 'u.' in the examples since they are methods of the object 'Unit u'. You should follow the patterns of the examples.

5. Write only the pseudocode inside '$< strategy >< /strategy >$' tag.

6. Do not write unnecessary symbols of the CFG such as, '$S \rightarrow$', '$\rightarrow$', etc.

7. Check the program and ensure it does not violate the rules of the CFG or the guidelines for writing the strategy.

8. The for loops in this language iterate over all units and the instructions inside the for loops attempt to assign actions to each of these units. That is why having nested for loops allows for a prioritization scheme. The innermost for-loops will contain the actions with the highest priority. Effective programs usually have nested for-loops.

## L.2 ENVIRONMENT DETAILS

NOWHERETORUN (9x8)

Consider a 9x8 gridded map of microRTS, a real-time strategy game. Consider this map as a 2-dimensional array with the following structure:

– There are a total of 8 neutral resource cells situated along the central column of the map, dividing the map into two parts. Each resource cell contains 10 units of resources.

– The base B1 of player 1 is located at index (1,1), which is located on the left side of the map.

– The base B2 of player 2 is located at index (7,6), which is located on the right side of the map.

– Each player controls one base, which initially has 5 units of resources.

– The only unit a player controls at the beginning of the game is the base.

DOUBLEGAME (24x24)

Consider a 24x24 gridded map of microRTS, a real-time strategy game. Consider this map as a 2-dimensional array with the following structure:

– There is a wall in the middle of the map consisting of two columns that has a small passage of 4 cells. The small passage consists of 4 resource cells each having only 1 resource.

– There are 28 resource cells at the top-left, top-right, bottom-left and bottom-right corners of the map respectively where each of them contains 10 units of resources.

– The bases of player 1 are located at indices (3,2) and (20,2), located on both sides of the wall.

– The bases of player 2 are located at indices (20,21) and (3,21), also located on both sides of the wall.

– Each player controls two bases, which initially have 5 units of resources each.

– There are 2 workers beside each base. So a total of 4 workers for each of the players.

BWDISTANTRESOURCES (32x32)

Consider a 32x32 map of microRTS, a real-time strategy game. Consider this map as a 2-dimensional array with the following structure:

– There are two L-shaped obstacles on the map, each with a passage of 4 cells located at the middle of left and right sides.

– There are a total of 12 neutral resource cells R located at the top-right and bottom-left corners of the map. Each resource center contains 20 units of resources.

– The base B1 of player 1 is located at index (6,14), which is located on the left side of the map.

– The base B2 of player 2 is located at index (25,17), which is located on the right side of the map.

– Each player controls one Base, which initially has 20 units of resources.

– There is one worker for each player besides their bases.

BLOODBATH (64x64)

> Consider a 64x64 gridded map of microRTS, a real-time strategy game. Consider this map as a 2-dimensional array with the following structure:
>
> – There are total 4 neutral resource cells situated close to the top-left, top-right, bottom-left and bottom-right sides of the map respectively. Each resource cell contains 40 units of resources.
>
> – There are obstacles in between each of the 4 resource centers.
>
> – The base B1 of player 1 is located at index (53, 55), which is located on the bottom-right side of the map.
>
> – The base B2 of player 2 is located at index (2, 6), which is located on the top-left side of the map.
>
> – Each player controls one Base each, which initially has 5 units of resources.
>
> – There is no worker for both player 1 and 2 in the initial map setup.
>
> – The only unit a player controls at the beginning of the game is the Base.

## M  SELF-PLAY LEARNING ALGORITHMS

Self-play algorithms attempt to approximate an optimal policy for two-player games. Iterated Best Response (IBR) (Lanctot et al., 2017) is perhaps the simplest self-play algorithm that we could use. IBR starts with an arbitrary policy $\pi_0$ in $[\![D]\!]$ for one of the players and approximates a best response $\pi_1$ to $\pi_0$ for the other player. Then it approximates a best response $\pi_2$ to $\pi_1$ for the first player. This process is repeated a number of times $m$, which is normally determined by a computational budget. The last resulting policy $\pi_m$ is returned as IBR's approximate optimal policy for the game.

The self-play process IBR follows generates a sequence of policies for the players, but IBR only considers the latest policy while computing a best response. Other algorithms, such as Fictitious Play (FP) (Brown, 1951), compute best responses to a policy that mixes the best responses computed in all previous iterations. The use of more policies allows the method to find optimal policies even in games with cyclic dynamics such as Rock, Paper, and Scissors.

The algorithm we use in our experiments, Local Learner (2L) (Moraes et al., 2023), also considers multiple policies in its self-play loop. This allows 2L to use more information than IBR. However, instead of including all policies seen in the process, like FP does, it selects a subset of the policies seen in the process. Using all policies can be computationally wasteful, as many of the policies are "redundant". 2L selects which policies to use based on the information collected during the search for programmatic policies. We refer the reader to the work of Moraes et al. (2023) for a detailed explanation of 2L.

