# OpenReview forum: "InnateCoder: Learning Programmatic Options with Foundation Models"
_ICLR.cc/2025/Conference — Submitted to ICLR 2025_

### Official Review · Reviewer_qHn7 · 2024-10-28

**Soundness:** 3
**Presentation:** 3
**Contribution:** 3
**Rating:** 6
**Confidence:** 4

**Summary:**

This paper presents a novel framework called InnateCoder which leverages foundation models (FM) and stochastic hill-climbing (SHC) to search programmatic policies to solve various RL problems. InnateCoder encode general human knowledge from foundation models in a zero-shot setting to formulate programmactic policies, while these policies interact with RL environment using options as "innate skills". The framework works as follows: given the environment description and its corresponding domain specific language (DSL), a prompt can be built to query the foundation models to generate programmatic policies. Although the generated programs are unlikely to solve the problem directly, they provide important corpus of options. By tweaking the syntax and semantics of these options, new options and programs are obtained for the framework to evaluate and search via SHC. The search stops when run out of budgets or the optimal solution is found.

**Strengths:**

- Presents a novel method called InnateCoder which leverages foundation models to provide corpus of knowledge instead of trying to solve the problem directly. Authors claim that the generated programmatic policies can be used as the starting point of option and program searching, and InnateCoder should extend the searching by extending to and visiting neighborhood programs. This method leverages the foundation models' great ability on higher-level abstraction and left the lower-level operations to options. The overall method is simple and very intuitive.
- Performes extensive evaluations on different benchmarks. The baselines are properly selected. The proposed framework is highly competitive in nearly all evaluated benchmarks.
- Related disccusion is extensive and informative. The discussion is helful and meaningful for researchers in related community.

**Weaknesses:**

- The main text does not provide a single complete example programmatic policies or an option for a single environment, it could be helpful to add at least one for better readability.
- The discussion on $\epsilon$ in SHC is lacking. In Sec. 4.1 line 371, it says the results shows the semantic space can be less conducive to search than the syntax space, depinding on the quality of the options used to induce it. If so, modifying the $\epsilon$ may improve the results.
- This is minor. The Sec. 3.1 introduces the definition of options. I think it might be better to introduce this concept informally before the formal definition.

**Questions:**

- In Sec. 4.2, why the results of Llama 3.1 + GPT-40 in Mayari is worse than the results of Llama 3.1 or GPT-4o alone?
- I understand how does the filtering options work in  Sec. 3.2.1 in general. However, could you demonstrate some specific numbers if possible?  For example, people would like to see the percentage of adopted options for each environment.
- The SHC looks very nice in many domains. Could you briefly summarize how does it work, e.g., how does SHC accquire the inital programmatic component? By enumeration of the DSL?
- I did not read the prompt line by line, so sorry if I miss anything. When you use prompts to query the programmatic policies, do you provide any multi-modal information other than plain text, such as images or animations? If not, do you think it is possible to improve the quality of the generated programs and options?

---

> ### Author Response · Authors · 2024-11-15
> **Thank you**
>
> **Can you provide examples of the programs?**
>
> **Answer:** Thank you for suggesting this. Please see Appendix E (Figure 10) of the revised version for a representative example of a final policy and an example of a program a foundation model generates.
>
> **A discussion on epsilon is lacking.**
>
> **Answer:** We added a plot in Appendix G (Figure 12) of the revised version where we show how the performance of InnateCoder changes as we vary the value of epsilon. What we observe is that InnateCoder is robust to the value of epsilon, as it presents similar performance for a wide range of values. We also observe that values that are “too small” or “too large” perform worse than those that find a good balance between the syntax (discovering new programs) and semantics (using existing programs).
>
> **This is minor. The Sec. 3.1 introduces the definition of options. I think it might be better to introduce this concept informally before the formal definition.**
>
> **Answer:** We have added a few intuitive sentences in the beginning of Section 3.1.
>
> **Why are the results of Llama 3.1 + GPT-4o in Mayari worse than the results of Llama 3.1 or GPT-4o alone?**
>
> **Answer:** The combination of programs written by Llama 3.1 and GPT-4o does not lead to “monotonic improvements”, as evidenced by the drop in performance against Mayari. This happens because none of the competition winners is constrained by the DSL we use in our experiments. As a result, the optimization done in self-play might not be specific for the opponents evaluated in Table 1, but to policies written in the DSL and encountered during the self-play process.
>
> We added the above paragraph in Section 4.2 of the revised version.
>
> **What is the percentage of options adopted?**
>
> **Answer:** Great question! Following your suggestion, we added an example in Appendix F of the revised version where we show that options can cover a large number of lines in the policy that InnateCoder returns. Importantly, we also show that on average 63% of the options are used in a best response during self play. Please see Appendix F for details.
>
> **SHC looks quite nice, can you summarize it?**
>
> **Answer:** Yes, SHC is a strong baseline. The initial program is obtained by sampling the production rules of the grammar that defines the DSL. The structure of the underlying space is given by a neighborhood function, which essentially mutates the tree that represents a program. Please see Definition 1 and the two paragraphs that follow the definition for a more detailed explanation.
>
> **How about multi-modal information as input?**
>
> **Answer:** We only use text, but we believe the system could be improved by using multimodal inputs, as suggested. This is an exciting direction for future research.

---

> > ### Comment · Reviewer_qHn7 · 2024-11-16
> >
> > Thanks for your comment as it addressed most of my concerns. I will keep my score and happy to see if it got accepted. In the future revision of the paper, make sure to increase the quality of the figures and try to make them consistent.

---

> > > ### Author Response · Authors · 2024-11-18
> > >
> > > Thank you for taking the time to review our rebuttal. We appreciate your support and your suggestion regarding the figures. In an eventual camera-ready revision, we will ensure that the figures are consistent in size and quality.

---

### Official Review · Reviewer_nF6R · 2024-10-29

**Soundness:** 3
**Presentation:** 2
**Contribution:** 4
**Rating:** 6
**Confidence:** 3

**Summary:**

This work introduces a new approach for learning policies for reinforcement learning tasks, by using the knowledge from pretrained LLMs. Instead of learning a NN policy, as is common in regular DRL, InnateCoder generates a number of options, each represented as a program in a custom DSL. These are then combined into a policy, with the exact combination being obtained via stochastic hill climbing during the training process.

**Strengths:**

This paper works on a very interesting direction of using pretrained LLMs to synthesize agent policies in RL tasks. This is important, as we still lack any meaningful foundation models for agents (here understood as action-taking systems), and this seems to be one good way of using the "general knowledge machines" for some agentic behavior.

The results obtained with this approach seem very good, and claim SOTA - I'm not sufficiently familiar with this specific domain to evaluate this claim, but it seems reasonable as it's compared to winners from public competitions.

**Weaknesses:**

My only main concern is about the fairness of comparison in the MicroRTS evaluation, possibly due to not understanding this part of the paper. In Figure 3, you plot the winrate. The winrate is defined on line 341 (somewhat confusingly under "Other specifications") as "The winning rate of a policy is computed for a set of opponent policies [...]" - what policies? Is it equally sampled between COAC, Mayari and RAISocketAI?

More importantly, over the course of the training in Figure 3 (the one that goes up to 1e5 games), is each method exclusively trained on self-play, and then checkpoints are "separately" evaluated against reference opponent policies (COAC, Mayari, RAISocketAI presumably)? Or do they interact with those policies throughout the training process?

Generally speaking, the writing is at times difficult to follow, possibly because I'm not that familiar with this specific subfield. For example, in line 283, "2L" is introduced as the algorithm used to train the policies, without much further elaboration or expanding the acronym. There is, of course, a reference to the paper that introduced it, but seeing as it's not a widely known algorithm, it might be useful to include some outline of how it works.

As a half-nitpick: the pixelated aesthetic of Figure 4 makes it rather difficult to tell what's going on. Consider for example FourCorners, where I can see something happening early on in the training, but mostly it's just straight converged lines. Or OneStroke, where it's just the straight converged lines. I understand it's difficult to present so much information in a finite amount of space, but perhaps there's some other way to make it work?

Furthermore, I'd be curious to see how well regular DRL algorithms would perform on Karel as a baseline - it does not seem to be an overwhelmingly difficult task, though I have not used it personally.

**Questions:**

Can you elaborate on the exact training procedure? Specifically - when training e.g. IC-Llama, what exactly is the workflow of the policy being improved, the round-robin tournament between 30 seeds, what data is used to improve the policy, and how/when it's evaluated?

My only real doubt is regarding the fairness of training IC against fixed versions of opponent policies. Also, fairness of the evaluations - how do the winrates differ against different reference policies? (for example, it could be the case that IC does really well against weaker opponents, but still mostly loses to the strongest policy, but all of this gets averaged to a high winrate on the graph - I don't claim that this is happening, but I'm not confident that it's not happening given the data in the paper)

I'm looking forward to seeing this cleared up during the discussion, and if by the end of it I'm fully convinced that this method actually outperforms all other known alternatives on these benchmarks (mainly MicroRTS, but also Karel - although in the latter I suppose SOTA isn't as difficult as in a competitive game), I'd be happy to increase my rating.

---

> ### Author Response · Authors · 2024-11-15
> **Thank you**
>
> **Fair comparison in MicroRTS. Does InnateCoder have access to the policies it is evaluated against during training?**
>
> **Answer:** No, InnateCoder does not have access to the policies it is evaluated against during training. It can only see policies that are generated in the self play process. Once the self-play process is finished, we evaluate the generated policies against held-out policies.
> Note that the results in Figure 3 are not computed with respect to the winners of the competition. They are computed with respect to the last policy each of the following evaluated methods generate: FM, IC-GPT, IC-Llama, LISS-o, LISS-r, SHC. Each of these methods is trained in self-play and, importantly, they do not “interact” with policies of other methods during training.
> We clarified the empirical design in the revised version by splitting the experiments paragraph into three parts: Efficiency Experiment (Figure 3), Competition Experiment (Table 1), and Size and Information Experiments (Figures 5 and 8). Please let us know if this split still does not clarify the empirical setup.
>
> **Could you add an explanation of 2L?**
>
> **Answer:** Thank you for the suggestion. Please see a brief description of 2L in Appendix M.
>
> **Figure 4 is pixelated.**
>
> **Answer:** We tried to improve the quality of the plots; please let us know what you think of them (Figure 4 of the revised version).
>
> **How well would Deep RL perform on Karel?**
>
> **Answer:** We evaluated the state-of-the-art DRL method for Karel, HPRL-PPO [1], in Appendix B.2 (Table 2). InnateCoder is never worse than HPRL-PPO and often substantially superior.
>
> [1] Guan-Ting Liu, En-Pei Hu, Pu-Jen Cheng, Hung-Yi Lee, and Shao-Hua Sun. Hierarchical programmatic reinforcement learning via learning to compose programs. In Proceedings of the International Conference on Machine Learning, 2023.
>
> **Can you elaborate on the exact training procedure? What is the experimental design?**
>
> **Answer:** InnateCoder uses a foundation model to generate a set of programs that are broken down into options and used to induce the language’s underlying semantic space. IC then uses a self-play algorithm to produce one program for playing on that map. We repeat this process 30 times to generate 30 agents.
>
> In Figure 3, for each of the 30 seeds, we evaluate the program a method generates after X games played against the last program the methods evaluated generate. That way we compute the values for different values of X (x-axis). Note this is the evaluation of the methods, not their training. During training they only have access to the programs generated through self-play. What you called reference policies are the policies the evaluated methods generate after they have exhausted their computational budget.
>
> As for the evaluation against competition bots (Table 1), we randomly select 9 out of the 30 programs from the previous experiment and evaluate them against the 3 winners of the last competitions on the 4 maps we use in our experiments. Table 1 presents the average results of 2 matches for each map, for a total of 2 * 4 * 9 = 72 matches. It is remarkable that the policies InnateCoder generates, which were not tuned to defeat the competition agents, can have such a high average winning rate: larger than 50% in all three tests.
> Note that our initial description of how the numbers in Table 1 was computed was incorrect; this is now fixed in the revised version. Thank you for asking this question.
>
> **I'm looking forward to seeing this cleared up during the discussion, and if by the end of it I'm fully convinced that this method actually outperforms all other known alternatives on these benchmarks (mainly MicroRTS, but also Karel - although in the latter I suppose SOTA isn't as difficult as in a competitive game), I'd be happy to increase my rating.**
>
> **Answer:** We hope our answers above clarify your doubts, and that now you understand that InnateCoder presents SOTA results in both Karel and MicroRTS. We also hope the new organization of the empirical section will make it easier for other readers to understand our experimental design. Please let us know if you have any other questions and/or suggestions.

---

> ### Author Response · Authors · 2024-11-25
>
> Thank you again for writing a thoughtful review on our work. Since the deadline for interacting with authors finishes tomorrow (November 26), we were wondering if you had the chance to read our rebuttal and if you have any follow-up questions.
>
> In particular, hopefully our response was clear in explaining that InnateCoder presented SOTA results on both Karel and MicroRTS.

---

### Official Review · Reviewer_gPM1 · 2024-11-04

**Soundness:** 3
**Presentation:** 3
**Contribution:** 2
**Rating:** 6
**Confidence:** 3

**Summary:**

Authors introduce a new way to learn programmatic options.  The set of options is built by first sampling a set of programs from a foundation model that is given a description of the MDP and the domain-specific language (DSL).  Then the set of programs is improved by searching through sets of modified programs to find ones with high reward.  Results show that this approach outperforms (i) other programmatic search algorithms that do not use foundation models and (ii) Deep RL approaches in the games MicroRTS and Karel the Robot.

**Strengths:**

- Authors do a nice job of explaining the algorithm.
- Authors also did a nice job of providing the DSL for each environment and the exact prompts used.

**Weaknesses:**

The main weakness here is that the contribution seems to be small.  It seems likely that the better use of a LLM is to learn a policy that outputs short programs conditioned on the history of the game that has occurred.  This way the foundation model can adjust its output as it gets more information on how the game works.  The reasoning provided for not testing this is that this would be expensive.  But if you were to apply this algorithm from scratch to a new environment, it would already be costly to come up with a sufficiently expressive DSL.

It is also unclear to me why machine learning is needing for settings like these.  The hand-engineered action space of (non-terminals, terminals, etc.) is already small, so regular software engineering should often be able to learn the sequence of these actions.

In addition, although it was helpful to see the exact prompts, the authors could be more transparent of how much prompt engineering was needed.  The exact "Program writing guidelines" and the "list of tasks" provided make it seem like there may have been quite a bit of prompt tuning was needed.

**Questions:**

1.  Can you discuss the types of prompt engineering that was required?
2.  Can you provide some examples of what the final programs look like in these domains?  It is difficult to know how complex these domains are.
3.  Is there any results on what type of search is more important between syntax and semantic search?

I am willing to increase my score if I can get a better sense of the value add of using machine learning here vs. just hand-crafting policies.

---

> ### Author Response · Authors · 2024-11-15
> **Thank you**
>
> **Creating a DSL is costly, why not learn a policy that outputs programs?**
>
> **Answer:** We recognize that writing a DSL from scratch is costly, but it is a cost that occurs only once. After a DSL is created, it can be used to solve as many problems as needed, which amortizes the cost of creating it. Take for example the DSL used in Microsoft Excel's FlashFill. It was designed once and it is used daily by millions of people around the world. By contrast, the use of an LLM to generate the programs and effectively guide the search must be paid for every problem one wants to solve. Having said that, the idea of training a model to output programs based on the game history is also interesting and we hope the community will investigate it in future works.
>
> **Why is ML needed for this setting?**
>
> **Answer:** Recently, a software engineer from a giant Computer Games company personally reported to us that they can spend months writing programs encoding agent behavior in video games. Our methods could be used to automate or semi-automate the process of writing these programs, potentially saving hundreds of hours of professional programmers.
> Moreover, previous work compared the programs professional programmers wrote with those one of the baselines we used in our experiments wrote (SHC) [1]. They showed that a version of SHC can write stronger programs than the human programmers enlisted for their experiment.
>
> [1] Programmatic Strategies for Real-Time Strategy Games. Julian Mariño, Rubens Moraes, Tassiana Oliveira, Claudio Toledo, and Levi Lelis. In the Proceedings of the Conference on Artificial Intelligence (AAAI), 2021.
>
> **I am willing to increase my score if I can get a better sense of the value add of using machine learning here vs. just hand-crafting policies.**
>
> **Answer:** Thank you for giving us the chance to explain the importance of machine learning in generating programmatic policies. Please do not hesitate to ask follow-up questions.
>
> **How much prompt engineering was needed?**
>
> **Answer:** All of our prompt engineering work was to ensure that the LLM could produce programs in a language that it was not trained on, and we did not spend more than a couple of days writing the prompts. Note that there are other alternatives to prompt engineering that could produce similar or even better results such as fine-tuning and token masking (e.g., https://blog.dottxt.co/coding-for-structured-generation.html). Our work shows what is possible in terms of generating programmatic options, and there are many ways of improving our system. These are interesting directions for future research.
>
> **Can you provide examples of programs?**
>
> **Answer:** Thank you for suggesting this. Please see Appendix E (Figure 10) of the revised version for a representative example of a final policy and an example of a program a foundation model generates. We also explain parts of the example in Appendix E, to demonstrate that the programs generated can be fairly complex.
>
> **Which type of search is more important between syntax and semantic?**
>
> **Answer:** The two searches are equally important and must be used together. The search in the syntax space allows you to find novel programs, while the search in the semantic space allows you to combine existing programs. We added a plot conveying this intuition in Appendix G (Figure 12) of the revised version.

---

> ### Author Response · Authors · 2024-11-25
>
> Thank you again for writing a thoughtful review on our work. Since the deadline for interacting with authors finishes tomorrow (November 26), we were wondering if you had the chance to read our rebuttal and if you have any follow-up questions.
>
> In particular, hopefully our response was clear in explaining that ML can have an important impact in the problem of automatically writing programs that encode policies.

---

> > ### Comment · Reviewer_gPM1 · 2024-11-25
> >
> > Thank you authors for your response to my questions.  I have increased my score to above the acceptance threshold as I believe the paper presents a sufficiently new and principled approach.  I still strongly believe this way of training agents with programmatic options is not the right direction for creating general purpose agents due to the large amount of human engineering required.  But I do not like to reject papers for subjective reasons if they are sufficiently new and principled so I have raised my score.

---

> > > ### Author Response · Authors · 2024-11-27
> > >
> > > Thank you for reading our rebuttal and increasing your score. We are hoping to see less and less human engineering as we make progress in this area.

---

### Official Review · Reviewer_LYYo · 2024-11-04

**Soundness:** 3
**Presentation:** 2
**Contribution:** 2
**Rating:** 5
**Confidence:** 3

**Summary:**

InnateCoder uses foundations models to generate domain-specific language candidates (options), and then use stochastic hill-climbing in the induced syntax / semantics space to find a best-performing agent.

**Strengths:**

- Writing overall is fine, although I have some small complaints which I'll mention in the weakness section.
- I believe this method is novel to my knowledge.
- InnateCoder was evaluated in two domains used in previous methods, showing nontrivial improvement over baselines in a number of tasks while matching performances in the rest.
- I appreciate the authors' effort to eliminate data leakage as a factor during evaluation.

**Weaknesses:**

- I find the presentation of this paper to be a little confusing for someone not familiar with the prior work. For example, in section 2, the authors start to talk about the pros and cons of DSL before giving an introduction or even a definition of DSL. I also think using the actual language (or a simplified version) in one of the tested domains, instead of the generic "if b then c" or "c1" "c2" as the example could be more helpful.
- Could the authors specify what are the exact differences between InnateCoder and LISS? Is the option source the only difference?
- I have concerns that application of this method could be limited. The DSL-dependency demands all discrete actions with a hand-crafted grammar.
- I would suggest showing some examples of the foundation model / final policy (can put those in the appendix).

Minor issues:

- Figure 1 is after Figure 2.
- 3.2.1 is the only subsection under 3.2.
- L208 "starting at the non-terminal __ the node represents".
- L230 "that is is".

**Questions:**

- The policies produced by InnateCoder seem to be deterministic. Is this the case for other baselines? How are the numbers in Table 1 produced? Are they calculated (by enumerating through all initial positions) or simulated (for a certain number of games and taking the average)? If the latter, how many games did you run to get the numbers?
- Why is there a clear difference between agents with $\leq 1400$ options and with $\geq 5000$ options? Agents with $\geq 5000$ options have sharp performance increases in the early stage of learning while agents with $\leq 1400$ options don't and can even have performance drops. Despite having such a distinction between the two groups, the curves within the two groups look similar. What factors contribute to this cut-off?
- L210 "$\mathcal{E}$ is an approximation of ..., ${\hat{V}^n(s_0)}$". What is the difference between $\mathcal{E}$ and ${\hat{V}^n}$?

---

> ### Author Response · Authors · 2024-11-15
> **Thank you**
>
> **Issues with the presentation**
>
> **Answer:** Thank you for raising concerns about the presentation. We have removed the discussion around DSLs in the problem definition section, as it was redundant with the discussion we had in the introduction. We also replaced the abstract example with a simplified version of the DSL we use with Karel. Please see the chances in blue in the revised version (Figure 1 and Section 2) and let us know if you would like to see further changes.
>
> **Differences between InnateCoder and LISS**
>
> **Answer:** LISS can only be used in a transfer learning setting, where the agent learns by solving multiple environments. InnateCoder allows us to induce semantic spaces in a zero-shot setting. In addition to the source of options, we connect semantic spaces with the option framework in RL. This is interesting because it shows a novel way of using options that, for the first time, can benefit from thousands of options.
>
> **Concerns with the dependency on DSLs**
>
> **Answer:** The dependency on DSLs is a weakness of program synthesis approaches at large. As these methods improve, they will use general-purpose languages, thus bypassing the need of a DSL. If synthesizers were good enough to use general-purpose languages, InnateCoder could be used as is. In fact, due to current LLMs’ ability to write programs in general-purpose languages, InnateCoder's performance might improve once we consider general-purpose languages.
>
> **Show examples of programs**
>
> **Answer:** Thank you for suggesting this. Please see Appendix E (Figure 10) of the revised version for a representative example of a final policy and an example of a program a foundation model generates.
>
> **Are InnateCoder and baselines deterministic?**
>
> **Answer:** Yes, all programs written in the DSLs used in our experiments are deterministic. This is true for all baseline, including the Deep RL baselines (RAISocketAI in Table 1, DRL in Figures 6 and 7, and  HPRL-PPO in Table 2).
>
> **How are the numbers in Table 1 produced?**
> **Answer:** Thank you for asking this question. It made us notice that our description of how the numbers are computed for Table 1 was incorrect. This is fixed in the revised version of the paper (see the paragraph “Competition Experiment” in Section 4).
>
> Here is how the computation is done. We randomly select 9 out of the 30 programs generated in the “Efficiency Experiment” and each of these programs plays 2 matches against the 3 winners of the last competitions. Since we use 4 maps, we have a total of 4 * 2 * 9 = 72 matches. We present the average winning rate against each opponent. Recall that the winning rate for each match can be 0, 0.5 or 1.0, for loss, draw, and victory. This experiment measures the average winning rate of the programs InnateCoder generates against the best-known agents for MicroRTS.
>
> **Why is there a gap in performance between <=1400 and >=5000?**
>
> **Answer:** The gap in performance is due to the increased chance of sampling helpful options with larger sets. We added in Appendix F of the revised version an example that illustrates this. In the example, we show a program that uses several options that are present in the set of 5000 of a run of InnateCoder, but not in the set of 1400, of another run of the algorithm.
>
> **What is the difference between E and V^n?**
>
> **Answer:** E is defined as V^n(s0). The reason why we wrote it this way was to differentiate the search from the evaluation of our application domain. We would be happy to use V^n(s0) instead of E, if the reviewer believes that it would improve the readability of the paper.
>
> **Figure 1 is after Figure 2.**
> **Answer:** This was fixed in the revised version, as we modified Figure 1 following your suggestion.
>
> **3.2.1 is the only subsection under 3.2.**
> **Answer:** We removed that heading in the revised paper.
>
> **L208 "starting at the non-terminal __ the node represents".**
> **Answer:** Added the word “symbol”.
>
> **L230 "that is is".**
> **Answer:** Fixed.

---

> > ### Comment · Reviewer_LYYo · 2024-11-21
> >
> > Thank you for the response. Most of my concerns have been addressed. I am still unsure about the DSL dependency. The limitation for me is two-fold - we not only need to design a fully expressive DSL and the corresponding skills for each domain, but also assume that the tasks are indeed solvable with a DSL language. You pointed out that this is a general problem in the field beyond this work's scope. I personally cannot verify this since I am not that familiar with this particular subfield, but I wouldn't argue about it if other reviewers agree.

---

> > > ### Author Response · Authors · 2024-11-21
> > >
> > > Thank you for taking the time to read our rebuttal and to comment on it. We should have provided evidence in our rebuttal to the claim that designing DSLs is common practice in the field. Here is a non-exhaustive list of papers that design and use a DSL to solve control problems. We provide in the list below where in each paper the DSL is defined.
> > >
> > > Please do not hesitate to ask follow-up questions.
> > >
> > > 1. Abhinav Verma, Vijayaraghavan Murali, Rishabh Singh, Pushmeet Kohli, and Swarat Chaudhuri. Programmatically Interpretable Reinforcement Learning. International Conference on Machine Learning (ICML), 2018. **See Equation at the top of the second column on page 3 of the pdf.** (https://arxiv.org/pdf/1804.02477)
> > >
> > > 2. Abhinav Verma, Hoang M. Le, Yisong Yue, and Swarat Chaudhuri. Imitation-Projected Programmatic Reinforcement Learning. Neural Information Processing Systems (NeurIPS), 2019. **See Figure 1 on page 2.** (https://arxiv.org/pdf/1907.05431)
> > >
> > > 3. Jeevana Priya Inala, Osbert Bastani, Zenna Tavares, Armando Solar-Lezama, Synthesizing Programmatic Policies that Inductively Generalize, 8th International Conference on Learning Representations (ICLR) 2020. **See Equation at the top of page 5.** (https://openreview.net/pdf?id=S1l8oANFDH)
> > >
> > > 4. Dweep Trivedi, Jesse Zhang, Shao-Hua Sun, Joseph J. Lim. Learning to Synthesize Programs as Interpretable and Generalizable Policies. Neural Information Processing Systems (NeurIPS) 2021. **See Figure 1 on page 3** (https://arxiv.org/pdf/2108.13643)
> > >
> > > 5. Leandro Medeiros, David Aleixo, and Levi Lelis. What can we Learn Even From the Weakest? Learning Sketches for Programmatic Strategies. In the Proceedings of the Conference on Artificial Intelligence (AAAI), 2022. **See Equation under “Domain-Specific Language for Can't Stop” on page 9 and Equation under Domain-Specific Language for MicroRTS on page 10.** (https://arxiv.org/pdf/2203.11912)
> > >
> > > 6. Guan-Ting Liu, En-Pei Hu, Pu-Jen Cheng, Hung-Yi Lee, Shao-Hua Sun. Hierarchical Programmatic Reinforcement Learning via Learning to Compose Programs. International Conference on Machine Learning (ICML) 2023.
> > > **See Figure 1 on page 2**. (https://arxiv.org/abs/2301.12950)

---

> > > ### Author Response · Authors · 2024-11-27
> > >
> > > Thank you again for providing feedback on our work. We were wondering if you had the chance to review the list of papers we provided in our previous message, which highlights the use of DSLs in the literature. We were curious whether this list, along with the consensus of the other reviewers in recommending acceptance, might influence your evaluation of our submission.

---

### Author Response · Authors · 2024-11-15
**Summary of Revisions**

We would like to thank all reviewers for taking the time and providing invaluable feedback on our manuscript. We have edited the paper to deal with all suggestions and comments to the best of our abilities. We believe that the paper has improved substantially after the changes. We have uploaded a revised version of our manuscript where the edits are highlighted in blue. Please do not hesitate to ask any further questions, we would be happy to promptly answer them.

Here is a summary of the changes we made.

1. Changed Figure 1 to be a concrete example based on Karel the Robot (**Reviewer LYYo**).
2. Added an explanation in the second paragraph of Appendix F for the divide between versions of InnateCoder that use 1400 or fewer options and versions that use 5000 or more (**Reviewer LYYo**).
3. Provided in Appendix E (Figure 10) examples of the programs that InnateCoder generates (**Reviewers LYYo, gPM1, qHn7**).
4. Included in Appendix G (Figure 12) a study of how InnateCoder behaves for different values of epsilon (**Reviewers gPM1 and qHn7**).
5. Fixed the explanation of how the numbers in Table 1 are computed (**Reviewer nF6R**).
6. Explained in Section 4.2 why InnateCoder with a library generated with Llama 3.1 + GPT-4o performs worse against Mayari (**Reviewer qHn7**).
7. Added in Appendix F an example showing which parts of the final policy are composed of options from the initial library, and how often policies are used in training (**Reviewer qHn7**).
8. Answered all questions the reviewers asked (please see individual comments below).

We have moved parts of the Related Work section to Appendix A, so we could fit all the changes requested in the main paper. Please let us know if you feel strongly about this change.

---

### Meta-Review · Area_Chair_S5F3 · 2024-12-19

**Metareview:**

**Summary**: The paper introduces InnateCoder, a novel approach that leverages foundation models to generate programmatic options in a zero-shot setting for RL. InnateCoder uses these options to induce a semantic space and efficiently search for programmatic policies via a mixture of syntax and semantic neighborhoods. The proposed method is evaluated in two domains demonstrating improved sample efficiency and performance compared to baselines such as SHC and LISS variants. This work aims to address the slow learning process of RL agents by providing "innate skills" encoded in domain-specific languages (DSLs).

**Strengths**:
- Reviewers found the presented method to be novel, simple and intuitive.
- Some of the reviewers found the evaluation to be extensive, on different benchmarks, with properly-selected baselines.
- The method seems highly competitive in several tasks.

**Weaknesses**:
- Limited applicability: One of the reviewers has concerns that application of this method could be limited due to the dependence on hand-crafted DSLs. Another reviewer finds it unclear why machine learning is needed at all, since the method uses a small hand-engineered action space.
- Small contribution: Reviewer `gPM1` questioned whether the primary contribution lies in reusing existing methods (e.g., programmatic options, semantic spaces) rather than introducing fundamentally new techniques.
- Confusing presentation: Several reviewers complained about the writing being confusing, opaque or just not clear at times. The writing is at times hard to follow and the figures are not clear and could be improved. Although the rebuttal improved clarity, some sections, such as the differences between InnateCoder and LISS, remain underexplained.

**Recommendation**: Overall, this is a borderline case leaning toward rejection, as the paper demonstrates potential but lacks the novelty to stand out, and requires more significant improvements (e.g., broader applicability, reduced handcrafted DSL reliance) to justify inclusion in ICLR. Due to the apparent lack of enthusiasm in the reviewer pool (with all borderline takes), I am hesitant to accept this paper.

**Additional Comments On Reviewer Discussion:**

Reviewers appreciated the empirical rigor but remained divided on the paper’s novelty and broader applicability. The rebuttal improved clarity and addressed specific concerns, but foundational limitations (e.g., DSL dependency and incremental contribution) persist. Overall, while the paper demonstrates potential, it requires further refinement to meet ICLR’s standards for broader impact and applicability.

---

### Decision · Program_Chairs · 2025-01-22

Reject